# Knowledge Exchange with Confidence: Cost-Effective LLM Integration for Reliable and Efficient Visual Question Answering

**Mahsa Mozaffari** [1], **Hitesh Sapkota** [2]*, **Xumin Liu** [1], **Qi Yu** [1,2]*
[1] Rochester Institute of Technology
`mmozaffari@mail.rit.edu, {xumin.liu, qi.yu}@rit.edu`
[2] Amazon
`sapkoh@amazon.com`

## Abstract

Recent advances in large language models (LLMs) have improved the accuracy of visual question answering (VQA) systems. However, directly applying LLMs to VQA still presents several challenges: (a) suboptimal performance when handling questions from specialized domains, (b) higher computational costs and slower inference speed due to large model sizes, and (c) the absence of a systematic approach to precisely quantify the uncertainty of LLM responses, raising concerns about their reliability in high-stakes tasks. To address these issues, we propose an UNcertainty-aware LLM-Integrated VQA model (`Uni-VQA`). This model facilitates knowledge exchange between the LLM and a calibrated task-specific model (*i.e.,* `TS-VQA`), guided by reliable confidence scores, resulting in improved VQA accuracy, reliability, and inference speed. Our framework strategically leverages these confidence scores to manage the interaction between the LLM and `TS-VQA`: the specialized questions are answered by the `TS-VQA` model, while general knowledge questions are handled by the LLM. For questions requiring both specialized and general knowledge, the `TS-VQA` provides candidate answers, which the LLM then combines with its internal knowledge to generate a more accurate response. Extensive experiments on VQA datasets demonstrate the theoretically justified advantages of `Uni-VQA` over using the LLM or `TS-VQA` alone.

## 1 Introduction

Recent advances in Large Language Models (LLMs) have opened new opportunities to enhance Visual Question Answering (VQA) performance by leveraging the rich general knowledge these models acquire through large-scale pre-training. LLMs consistently achieve higher accuracy on VQA tasks compared to traditional task-specific VQA models (`TS-VQA`), which are smaller models trained specifically for visual question answering. However, fully relying on LLMs for VQA faces critical practical challenges that limit their real-world deployment.

The primary challenge is computational efficiency. LLMs typically require billions of parameters, resulting in prohibitive computational overhead, high financial costs, and significant inference latency. These limitations become critical in time-sensitive applications (Ding et al., 2025) and resource-constrained environments. Moreover, recent studies show that multi-purpose LLMs can be orders of magnitude more expensive to operate than task-specific models during inference. Environmental concerns add another layer of complexity, as large-scale models contribute substantially to carbon emissions and energy consumption (Strubell et al., 2020; Bommasani et al., 2021; Weidinger et al., 2022; Wu et al., 2022). Additionally, relying on third-party LLMs introduces recurring costs, and potential data privacy risks.

However, this computational burden may be unnecessary for many questions. Not all visual questions require the full power of massive language models – smaller `TS-VQA` models can effectively handle simpler queries while consuming significantly less computational resources. Furthermore,

---
*Work not related to Amazon.

Figure 1: Comparison of `TS-VQA` (VisualBERT), an LLM (Mistral-7B), and our hybrid `Uni-VQA` (73% delegation) in average latency, carbon emissions[1], accuracy, and ECE (Expected Calibration Error) on COCO-QA. (*ECE measures the gap between confidence and accuracy of the model; lower ECE is better.)

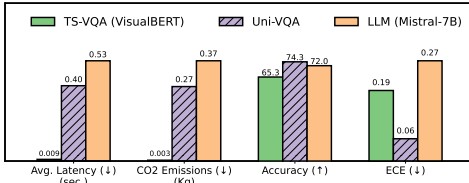

`TS-VQA` models trained on domain-specific data can sometimes provide more accurate answers than LLMs in specialized areas where the LLMs lack sufficient knowledge. Most importantly, our empirical analysis reveals that LLMs and `TS-VQA` models possess complementary strengths: even when `TS-VQA` models are uncertain about their final answers, they often generate valuable candidate answers that, when shared with LLMs, substantially improve LLM performance (as shown in Fig. 2a). This suggests an opportunity for a collaborative approach rather than simply choosing between the two model types.

Building on this observation, we propose a hybrid framework that enables strategic collaboration between `TS-VQA` models and LLMs. The key insight is to use the `TS-VQA` model's confidence scores to determine not only when to consult the LLM, but also when and how to transfer specialized knowledge from the `TS-VQA` model to enhance LLM reasoning. To realize this vision, we introduce `Uni-VQA` (UNcertainty-aware LLM integrated VQA), a novel hybrid framework that intelligently combines `TS-VQA` models with LLMs through confidence-based collaboration. However, standard VQA models trained with cross-entropy loss tend to be overconfident and poorly calibrated, as illustrated in Fig. 2b. They often produce incorrect answers with high confidence scores, meaning these scores cannot reliably indicate when the model is actually correct – rendering them untrustworthy for decision-making. To address this critical issue, we develop a calibration technique that ensures confidence scores accurately reflect the likelihood of correctness, as shown in Fig. 2c, enabling reliable confidence-based integration within our hybrid framework.

With properly calibrated confidence scores, `Uni-VQA` framework operates through a three-tiered knowledge exchange mechanism based on confidence levels. For highly specialized questions where the `TS-VQA` model exhibits high confidence (meaning answer is very likely to be correct), the system provides answers directly without consulting the LLM, leveraging the model's domain expertise efficiently. For questions requiring broad general knowledge, where the calibrated `TS-VQA` model shows low confidence, the framework delegates entirely to the LLM. Most importantly, for questions requiring both specialized and general knowledge – where the `TS-VQA` model has partial knowledge but remains uncertain – our framework enables a novel form of collaboration. The `TS-VQA` model transfers its specialized knowledge by providing dynamically selected candidate answers to the LLM, which the LLM then incorporates with its general knowledge to produce more accurate responses. This collaborative approach leverages the complementary strengths of both model types.

The overall framework during the inference is illustrated in Fig. 2d. By selectively delegating questions that require general knowledge to the LLM, our framework significantly reduces overall computation and inference costs, while achieving higher accuracy, compared to using the LLM alone, as shown in Fig. 1. Experimental results across multiple VQA datasets show that `Uni-VQA` outperforms both the LLM and `TS-VQA` used in isolation, while dramatically reducing computational overhead. Our contributions are summarized as follows:

- We develop a calibration technique that enables `TS-VQA` models to provide reliable confidence estimates essential for effective confidence-based hybrid integration.
- We introduce UNcertainty-aware LLM integrated VQA model (`Uni-VQA`) that enables cost-effective knowledge exchange between LLMs and calibrated `TS-VQA` models, improving both accuracy and efficiency through strategic collaboration.
- We provide a theoretical analysis to justify the calibration benefits of our diverse ensemble, and demonstrate substantial improvements in both accuracy and efficiency across multiple `TS-VQA` models and VQA datasets.

## 2 RELATED WORK

**Visual Question Answering.** To tackle the complex VQA problem, various methodologies have been developed (Schwenk et al., 2022; Lin et al., 2022; Gao et al., 2022; Qian et al., 2022). To

---

[1]Carbon emissions of models are estimated using `https://github.com/mlco2/codecarbon`.

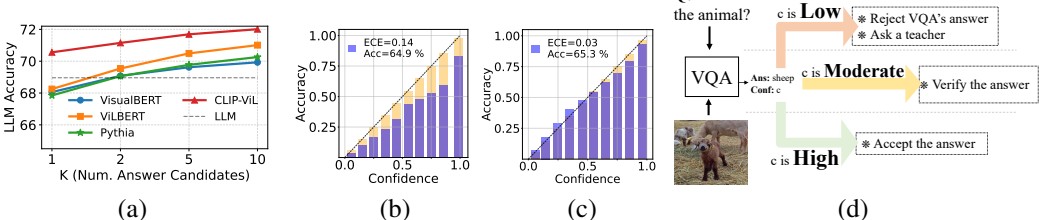

Figure 2: (a) Effectiveness of incorporating candidate answers of four `TS-VQA` models on the performance of Mistral-7B, demonstrating improved accuracy as more knowledge is shared with the LLM. (b) and (c) present reliability diagrams of a baseline and calibrated `TS-VQAs` (VisualBERT), respectively, showing how model confidence aligns with actual accuracy (orange bars represent a perfect calibration). (d) General workflow of `Uni-VQA` during inference, illustrating how confidence levels determine whether to use the VQA model directly, consult an LLM, or rely entirely on the LLM.

enhance the understanding of the context present in the VQA text, attention-based mechanisms have been used (Gao et al., 2019; Lu et al., 2018; Yu et al., 2019). Pre-training has also been leveraged, where models are first pre-trained using unlabeled data and then fine-tuned in the downstream VQA tasks (Shen et al., 2021; Alayrac et al., 2022; Zeng et al., 2023; Bao et al., 2022; Wang et al., 2023).

**Large Language Model-based VQA.** Due to pre-training and reasoning capabilities of LLMs, these models provide an implicit knowledge source for the VQA tasks. Yang et al. (2022) use image captions to provide visual context to GPT-3 as an implicit knowledge base for knowledge-based VQA task. Yu et al. (2023) propose a framework that prompts LLMs with complementary answer candidates and answer-aware examples to enhance OK-VQA performance. However, these LLM-based VQA models are inadequate for building reliable, efficient, and cost-effective VQA due to their total reliance on LLMs to address all the questions. The knowledge exchange between the LLM and the `TS-VQA` is not properly guided, which may lead to sub-optimal performance.

**Calibration in VQA.** Whitehead et al. (2022) formulate reliable VQA as selective prediction, where a model abstains on low-confidence answers using an additional selector to estimate confidence score. Dancette et al. (2023) reduce the need for extra held-out data by training the VQA model and selector jointly with peer-generated pseudo-labels. While these methods improve reliability through abstention, they do not directly address poor calibration and overconfidence. Complementary to selective prediction methods, GLEN directly addresses calibration in VQA by combining low-rank network factorization with a generalized focal-loss ensemble to reduce overconfident incorrect predictions while preserving VQA accuracy (Mozaffari et al., 2025). Uni-VQA instead uses calibrated confidence to guide LLM collaboration and candidate-answer exchange.

**Retrieval-Augmented Generation.** Our framework shares a high-level principle with Retrieval-Augmented Generation (RAG) methods (Guu et al., 2020; Lewis et al., 2020; Hu et al., 2023; Izacard & Grave, 2021): both augment LLMs with external modules. However, they target different bottlenecks and are complementary. RAG retrieves textual evidence from external corpora (e.g., the web, knowledge bases) to expand LLM knowledge coverage, typically invoking the LLM on every query. In contrast, `Uni-VQA` employs a calibrated `TS-VQA` model that provides candidate answers and confidence scores, enabling selective LLM invocation only for low- and mid-confidence cases. The two frameworks address orthogonal concerns: RAG controls what textual evidence the LLM sees; `Uni-VQA` controls when and how the LLM is used. Crucially, our diverse ensemble calibration naturally pushes out-of-distribution and knowledge-intensive questions (where `TS-VQA` lacks expertise), toward the lowest-confidence region (further discussion in Appendix G.11). This creates a natural integration point: questions routed to the LLM without `TS-VQA` candidates are those that would benefit most from RAG augmentation. Thus, `Uni-VQA` could directly wrap a RAG-enhanced LLM for low-confidence queries without modifying the calibration or routing logic.

## 3 METHODOLOGY

Assume $\mathcal{D}_N = \{(\mathbf{v}_n, \mathbf{q}_n, \mathbf{a}_n)\}_{n=1}^N$ is a dataset consisting of N instances, where each instance comprises an image $\mathbf{v}_n$, a question $\mathbf{q}_n$, and an answer $\mathbf{a}_n$. We establish $\mathcal{X} \equiv \mathcal{V} \times \mathcal{Q}$ as the input space, with $\mathbf{x}_n = (\mathbf{v}_n, \mathbf{q}_n)$ representing an input data point. Additional concepts utilized in the paper are elaborated in the Appendix.

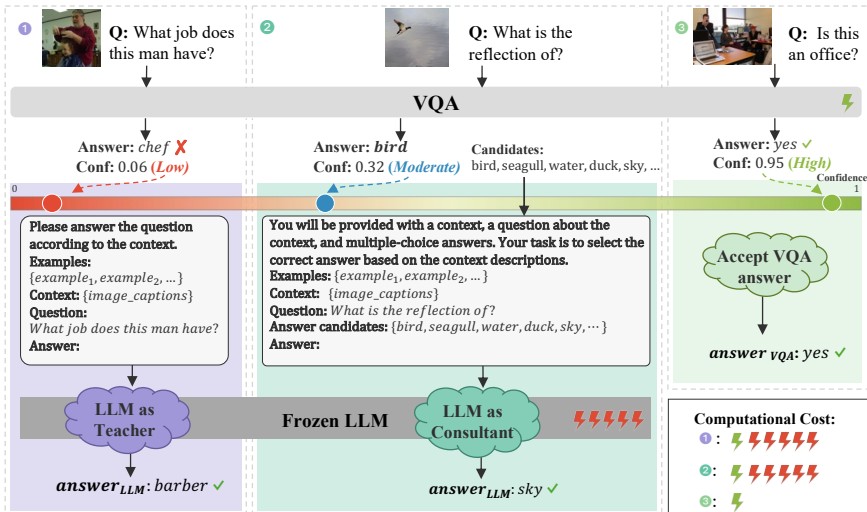

Figure 3: General overview of `Uni-VQA` framework at inference time. LLM serves different roles depending on the VQA's confidence. (a) If the VQA model is least confident, the LLM serves as a teacher and provides the answer to the question, (b) If VQA model is confused among multiple candidate answers, the LLM serves as a consultant and helps to select answer from candidate answers, (c) if VQA model is highly confident, `Uni-VQA` directly answers without LLM involvement.

## 3.1 OVERVIEW OF THE FRAMEWORK

Fig. 3 illustrates the overview of the proposed `Uni-VQA` framework. During the training phase, we first train a well-calibrated `TS-VQA` model employing a diverse ensemble based approach. This calibration step is crucial because reliable confidence scores enable effective integration between the `TS-VQA` and LLM components during inference. The inference phase operates through a confidence-guided process: Initially, the calibrated `TS-VQA` model generates an initial answer along with its associated confidence score $c$. Based on this confidence score, the framework routes the query to one of three distinct scenarios defined by confidence thresholds $l$ and $u$. 1) when the `TS-VQA` exhibits high confidence ($c \geq u$), the `TS-VQA` answer is accepted directly without LLM's involvement, leveraging the model's domain-specific expertise efficiently. 2) When confidence is low ($c < l$, typically for questions requiring broad general knowledge beyond `TS-VQA`'s specialization), the question is fully delegated to the LLM without answer candidates, which we refer to as the **LLM as Teacher** scenario. 3) For moderate confidence levels ($l \leq c < u$), where `TS-VQA` has partial knowledge but remains uncertain, answer candidates of `TS-VQA` are dynamically selected and provided to the LLM in what we call the **LLM as Consultant** scenario, enabling a collaborative reasoning where the LLM integrates these specialized insights with its own general knowledge.

This confidence-guided delegation mechanism strategically leverages the complementary strengths of both model types. It utilizes the `TS-VQA`'s domain-specific expertise with low computational cost for high-confidence questions, harnesses the LLM's broad reasoning capabilities for challenging general knowledge queries, and facilitates knowledge exchange through candidate answers when both specialized and general knowledge are needed to improve predictive performance.

## 3.2 RELIABLE VQA VIA MODEL CALIBRATION

In our `Uni-VQA` framework, the integration of LLM and `TS-VQA` models depends critically on the `TS-VQA` model's confidence estimates. For optimal integration, these confidence estimates must reliably indicate answer correctness, *i.e.,* low confidence should signal incorrect answers, while high confidence should signal correct ones. This requires well-calibrated `TS-VQA` models where confidence estimates accurately align with actual accuracies. Reliable uncertainty estimation, however, remains challenging across modern vision foundation models, vision-language models, and LLMs (Pandey et al., 2024; Tu et al., 2024; Xiong et al., 2024). In our setting, standard VQA models trained with cross-entropy loss suffer from overconfidence (Fig. 2b), consistently expressing higher confidence than their actual accuracies.

**Diverse Ensemble Strategy.** To address the calibration problem, we propose a Diverse Ensemble (DE) strategy that creates multiple complementary `TS-VQA` models, each specializing on different aspects of the data distribution. Deep ensembles are shown to effectively improve model accuracy and calibration (Wilson & Izmailov, 2020; Lakshminarayanan et al., 2017; Wood et al., 2023; Sapkota et al., 2023), particularly when diversity is enforced among base learners. Our approach leverages Distributionally Robust Optimization (DRO) (Duchi & Namkoong, 2019) to train an ensemble of $E$ diverse `TS-VQA` models that naturally complement each other.

Given training samples $\{x_n\}_{n=1}^N$ and per-sample loss $l(x_n, \Theta)$ (cross-entropy), we view calibration as learning under an adversarial reweighting of the empirical distribution. For each ensemble member, we minimize a DRO-style weighted loss:

$$\mathcal{L}_{DRO}(\Theta) = \sum_{n=1}^N \mathbf{w}_n l(\mathbf{x}_n, \Theta) \tag{1}$$

where $\mathbf{w}_n$ determines the emphasis on each training instance $\mathbf{x}_n$. The weight vector $\mathbf{w}$ is dynamically computed at every training step based on the current model's losses. Concretely, we adopt the regularized DRO formulation with KL-divergence, which yields the closed-form softmax weighting (see Appendix C for derivation):

$$w_n^*(\lambda) = \frac{\exp\big(l(x_n, \Theta)/\lambda\big)}{\sum_{j=1}^N \exp\big(l(x_j, \Theta)/\lambda\big)}.$$

where $\lambda > 0$ controls how far $\mathbf{w}^*$ can deviate from uniform weights, and thus how strongly the model focuses on high-loss (difficult) samples.

By varying the hyperparameter $\lambda$ across ensemble members, we obtain models that specialize on different difficulty regimes. When $\lambda$ is small, the weighting scheme emphasizes challenging samples, producing a model that tends to be cautious (lower confidence) since it has learned to handle difficult cases. When $\lambda$ is large, the weighting approaches uniform distribution, creating a model that captures general patterns and tends to be more confident on typical samples. In all experiments, we use an ensemble of $E = 3$ models with small, medium, and large $\lambda$ values, creating complementary expertise across the difficulty spectrum (See Appendix C and Appendix G.13 for details).

At inference time, we average the logits of the ensemble members, $f_{DE}(\mathbf{x}) = \frac{1}{E}\sum_{e=1}^E f_e(\mathbf{x})$, (where $f_e(\mathbf{x})$ represents the logits from the e-th ensemble member), and obtain the confidence score from the resulting softmax distribution. This combination naturally produces well-calibrated confidence scores because the cautious models (trained on hard samples) temper the overconfidence of models trained on easier samples, while confident predictions from easy-sample experts are validated across the ensemble.

As demonstrated in Figs. 11 and 12 in the appendix section G.3, our diverse ensemble significantly improves calibration by assigning appropriately lower confidence to incorrect answers while maintaining high confidence for correct responses, making the confidence scores a reliable indicator for our delegation mechanism.

### 3.3 CONFIDENCE GUIDED KNOWLEDGE EXCHANGE

Calibrated `TS-VQA` confidence scores are critical in selectively delegating challenging questions along with answer candidates to the LLM, not only improving the overall predictive performance but also enabling efficient inference of easier questions by the `TS-VQA` model. Additionally, the effectiveness of these candidate answers varies significantly across different confidence intervals.

Motivated by this observation, we hypothesize that across different confidence levels, the number of answer candidates the LLM benefits from varies by confidence level. Specifically, for an effective combination of LLM and VQA by answer-candidate augmentation, fewer answer-candidates are needed at high confidences of `TS-VQA`, while more candidates become beneficial as the confidences decrease. At lowest confidences, providing a large number of answer candidates is impractical, making it more effective for the LLM to answer the questions without relying on any answer candidates. To validate this hypothesis we compare the LLM's predictive accuracy within each confidence interval of the `TS-VQA` for varying number of answer candidates in top-0, top-1, top-2, and top-10 along with LLM's performance without answer candidates. As Fig. 4b suggests, in higher confidence intervals, LLM's performance is higher when fewer answer candidates are presented. As the

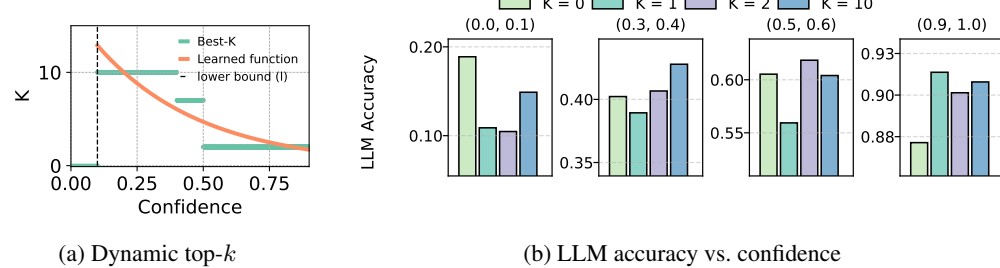

Figure 4: (a) Learned mapping: confidence to $k$. (b) Accuracy for $k \in \{0, 1, 2, 10\}$ across confidence bins.

confidence interval decreases, LLM's performance is enhanced when more answer candidates are included. In the lowest confidence intervals, the LLM's performance with answer candidates drops as compared to when no answer candidate is presented.

To that end, we propose a dynamic approach for effective answer candidate selection, informed by the TS-VQA's answer confidence. Considering $c_i = \max f_\Theta(\mathbf{x}_i)$ as the confidence of the predicted answer $\hat{a}_i$. We define $l$ as threshold for delegating to LLM with no answer candidates, and $u$ as thresholds for using TS-VQA for question answering. Specifically, if $c_i \geq u$, then the answer predicted by the TS-VQA model i.e. $\hat{a}_i$ is accepted, and if $c_i < l$, answering is delegated to LLM, without any answer candidates included in the prompt. For $u \geq c_i \geq l$, answering is delegated to LLM provided with $K(c_i) \geq 1$ answer-candidate where $K$ is determined by:

$$K(c_i) \approx \lceil M e^{-W(\frac{c_i - l}{u - l})} \rceil, \tag{2}$$

where $0 \leq l < u \leq 1$, and learnable parameters $M, W$ are determined based on a validation set and $\lceil x \rceil$ is the rounding operation that converts the fractional value into the closest integer. Fig. 4a presents the learned top-$k$ answer candidate selection for Calibrated CLIP-ViL.

### 3.4 ACCELERATING INFERENCE WITH KNOWLEDGE DISTILLATION

To further reduce the inference cost, we propose to leverage knowledge distillation to transfer the predictive accuracy and calibration of the diverse ensemble (DE) into a single TS-VQA model with the same architecture as the individual ensemble components. Instead of learning from target labels using cross-entropy loss, the distilled model minimizes the Kullback-Leibler divergence between its output distribution and the diverse ensemble's output logits distribution. This approach effectively preserves both accuracy and calibration with theoretical guarantees (Allen-Zhu & Li, 2023; Hebbalaguppe et al., 2024) while eliminating the additional computational burden of ensembling. Our experiments show that the distilled model maintains comparable ECE and accuracy (within 0.4%) while reducing latency by up to 60%. Further details and numerical results are provided in Appendix G.9.

## 4 THEORETICAL ANALYSIS

In this section, we theoretically demonstrate that the proposed Uni-VQA technique effectively delegates a greater number of incorrect predictions, that would otherwise be confidently wrong. We show this in two steps. First, we demonstrate how the diverse ensemble technique improves calibration. In the second step, we show that with better calibration, more incorrect samples are shifted into the low-confidence regions, allowing them to be effectively delegated to the LLM for correction. Complete proofs for the theoretical results are provided in Appendix D.

**Diverse ensemble improves the ECE.** In this section, we showcase the lemma demonstrating how diverse ensemble techniques improve the model calibration (i.e., ECE) compared to an Expected Risk Minimization (ERM)-based model. Specifically, in the following lemma, we show that the DE loss will be an upper bound on the regularized cross-entropy loss where the regularizer is the negative entropy of the predictive distribution $\hat{p} = f_\theta(\mathbf{x})$

**Lemma 4.1** *Consider $\mathcal{L}_{DE}(\theta)$ being the diverse ensemble loss and $\mathcal{L}_{CE}^e(\theta)$ being the cross entropy loss for the subnetwork $e$, and $\hat{p}^e = f_\theta^e(\mathbf{x})$ being the prediction distribution of the base subnetwork $e$. Then, we have:*

$$\mathcal{L}_{DE}(\theta) \geq \frac{1}{C} \sum_{e=1}^{|E|} [\mathcal{L}_{CE}^e(\theta) - \lambda_e \mathcal{H}^e[\hat{p}]] \tag{3}$$

where $|E|$ is the total number of subnetworks used in our ensemble, $C$ is the normalization constant of DRO weights, $\lambda_e$ is the DRO hyperparameter controlling the balance between CE loss and predictive entropy, and $\mathcal{H}^e[\hat{p}]$ being the entropy of the $\hat{p}$.

**Remark.** The Lemma indicates that minimizing the DE loss leads to: (a) minimization of the cross-entropy loss, and (b) an increase in the entropy of the predictive distribution $\hat{p}$. Increasing the entropy of the predictive distribution can avoid overconfident predictions produced by the neural network, thereby improving the calibration. As a result our approach will reduce the likelihood of errors in the high confidence region, ensuring that the incorrect predictions remain in the low confidence regions. These low-confidence questions are then delegated to the LLM, that provides the final answer with the support of the dynamically selected candidate answers from the `TS-VQA`.

**Diverse ensemble maximizes incorrect sample delegation.** Because of the improved calibration achieved through the diverse ensemble technique, our approach shifts more incorrect samples into the low confidence region compared to the ERM-based approach. This is because, ERM tends to produce overconfident prediction for most of the samples, causing many wrongly answered samples to fall in the high-confidence region (as shown empirically in Fig. 11). In contrast, diverse ensemble lowers confidence levels, leading to a higher number of samples in the low-confidence region.

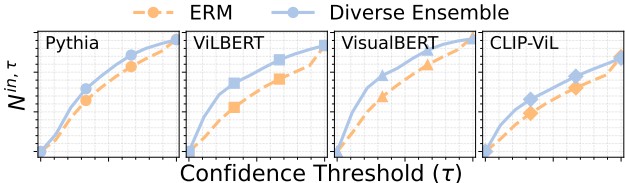

Figure 5: Empirical evidence illustrating $N_{\mathrm{DE}}^{in,\tau} \geq N_{\mathrm{ERM}}^{in,\tau}$, across four VQA architectures.

**Theorem 4.2** *Let $N_{DE}^{\tau}$ and $N_{ERM}^{\tau}$ be the total number of samples belonging to the low confidence region $\mathcal{R} : \{\hat{p} \in [0, ..., \tau]\}$ with $\tau$ being the threshold defining the low-confidence region. Then, for the region $\mathcal{R}$, the following holds true*

$$N_{DE}^{in,\tau} \geq N_{ERM}^{in,\tau} \tag{4}$$

*where $N_{DE}^{in,\tau}, N_{ERM}^{in,\tau}$ are the number of incorrect samples from DRO and ERM, respectively in region $\mathcal{R}$.*

**Remark.** By leveraging the DE-framework, we ensure that the incorrect samples are more likely to be in the low-confidence region, as empirically illustrated in Fig. 5. It maximizes the LLM's ability to correct these incorrect answers. In contrast, the ERM-based approach frequently assigns high confidence scores to incorrect samples due to overfitting. As a result, the delegation threshold must be set very high to pass these samples to the LLM for correction. This leads to either suboptimal accuracy if threshold is not high enough, or sub-optimal efficiency if the threshold is set too high, requiring more frequent delegation to the LLM.

## 5 EXPERIMENTS

We evaluated the performance of our `Uni-VQA` framework on multiple existing VQA architectures and report comparative quantitative results on the VQA-v2 (Antol et al., 2015) and COCO-QA (Ren et al., 2015) test splits, and conduct extensive ablation studies to justify the effectiveness of various proposed components. This includes effectiveness of (i) diverse ensemble-based VQA, (ii) answer-candidate augmented LLM prompting, and (iii) dynamic answer-candidate selection approach. Due to limited space, we have included more experimental results in the section G.9.

**Baselines.** We have considered five baselines. This includes (a) Pretrained-LLM, (b) `TS-VQA`, (c) VectorScale-based post-hoc calibrated VQA, referred to as VectorScale (Guo et al., 2017), (d) hybrid LLM-VQA confidence threshold based delegation, referred to as LLM-VQA, and (e) VQA models with our novel calibration technique denoted as `Calibrated`. LLM-VectorScale refers to integration of LLM with the VectorScale calibrated VQA. Specifically, in terms of VQA models, we consider five `TS-VQA` models: Pythia (Jiang et al., 2018), CLIP-ViL (Shen et al., 2021), ViLBERT (Lu et al., 2019), VisualBERT (Li et al., 2019), and BEiT-3 (Wang et al., 2023). Pythia (Jiang et al.,

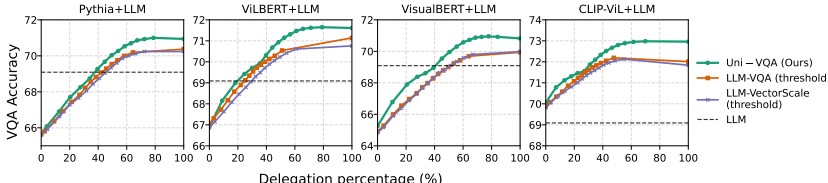

Figure 6: Performance comparison of the proposed method against LLM-VQA, and LLM-VectorScale (threshold), with respect to the delegation threshold. Accuracy at zero delegation is accuracy of `TS-VQA` model.

Figure 7: Performance comparison of the proposed method against LLM-VQA, and LLM-VectorScale (threshold), with fixed top-10 answer candidates, with respect to the delegation percentage.

2018) is a bottom-up top-down model, which leverages the up-down attention mechanism (Anderson et al., 2018), and combines the representations of question and image by element-wise multiplication. CLIP-ViL (Shen et al., 2021) uses the Movie-MCAN architecture (Nguyen et al., 2020) with the visual encoder of the CLIP (Radford et al., 2021) pre-training model. ViLBERT (Lu et al., 2019) and VisualBERT (Li et al., 2019) are pre-training-based transformer architectures with attention mechanisms. BEiT-3 is a state-of-the-art general-purpose vision-language model trained through masked-data modeling. For LLM-based models, we have employed frozen *Mistral-7B* (Jiang et al., 2023), and LLaVA-1.5 13B (Liu et al., 2023) as a VLLM.

**Dataset and evaluation metrics.** We use VQA-v2 (Antol et al., 2015) and COCO-QA (Ren et al., 2015) data sets. See section F.1 for more details. We utilize four metrics for evaluation: (1) VQA accuracy (`ACC`) to illustrate predictive performance, (2) Expected Calibration Error (`ECE`) which measures the difference between model confidence and actual accuracy (lower is better, with 0 indicating a perfect calibration), and is used to assess the reliability, (3) the proportion of questions assigned to LLM (`LLM-Deleg %`) as a proxy for computational expense and inference time, reflecting the extra cost incurred by LLM, and (4) Average latency of inference (`Latency`) measured in seconds. For complete implementation details refer to Appendix F.3.

## 5.1 COMPARISON RESULTS

Fig. 6 compares our approach with the baselines for various delegation thresholds. With the delegation threshold of 0, none of the questions is delegated to the LLM whereas, as the threshold increases, more questions with lower confidence scores are delegated to the LLM. LLM-only indicates the baseline result when we directly answer all questions using LLM. There are two key observations that can be inferred from Fig. 6. First, delegating low-confidence samples to the LLM improves performance across all baselines, including our `Uni-VQA`. This improvement can be attributed to the LLM's ability to handle challenging questions that the `TS-VQA` models struggle with. Second, due to its superior calibration, coupled with the uncertainty-sensitive dynamic delegation technique, our `Uni-VQA` delegates more incorrect samples to the LLM, achieving better overall performance compared to other baselines. This highlights the importance of calibration enhancement and dynamic delegation in hybrid VQA models.

Fig. 7 compares our `Uni-VQA` with baselines in terms of the VQA accuracy against the LLM-delegation percentage. First, our approach achieves the highest maximum accuracy among the baselines. At any given fixed delegation percentage, it also obtains a higher accuracy than the baselines. It's worth noting that, our model can match the accuracy of baselines with a lower delegation percentage, which implies a lower inference-time and computational overhead. For example, in VisualBERT, `Uni-VQA` achieves the same $68.3\%$ VQA accuracy as LLM-VectorScale but with $11.67\%$ lower LLM delegation. Table 1 further demonstrates the effectiveness of our `Uni-VQA` with regard to different VQA models against the competitive baselines. The Table mainly demonstrates two key phenomena. First, our calibration technique, **Calibrated** (Ours), improves the calibration performance i.e., ECE without compromising the accuracy. Second, due to the enhanced calibration, the presence of overconfident wrong predictions is effectively minimized in the highest confidence regions. As a result, the uncertainty-aware dynamic delegation ensures that easier questions—those

Table 1: Performance comparison of `Uni-VQA` with `TS-VQA` models and LLM across five architectures.

| Model | | VQA-v2 | | | | COCO-QA | | | |
|---|---|---|---|---|---|---|---|---|---|
| | | ACC↑ | ECE↓ | LLM-Deleg (%)↓ | Latency↓ | ACC↑ | ECE↓ | LLM-Deleg (%)↓ | Latency↓ |
| LLM-only (Mistral-7B) | | 69.09 | 0.31 | 100 | 0.534 | 72.03 | 0.27 | 100 | 0.534 |
| Pythia | Standard VQA | 65.67 | 0.14 | - | 0.003 | 68.62 | 0.16 | - | 0.003 |
| | VectorScale | 65.59 | 0.09 | - | 0.003 | 68.88 | 0.10 | - | 0.003 |
| | **Calibrated (Ours)** | 66.15 | 0.06 | - | 0.009 | 68.64 | **0.02** | - | 0.009 |
| | **Uni-VQA (Ours)** | **71.00** | **0.05** | 78.77 | 0.426 | **74.78** | 0.06 | 64.84 | 0.333 |
| CLIP-ViL | Standard VQA | 69.95 | 0.18 | - | 0.023 | 70.38 | 0.15 | - | 0.016 |
| | VectorScale | 69.81 | 0.15 | - | 0.031 | 70.41 | 0.11 | - | 0.017 |
| | **Calibrated (Ours)** | 70.05 | 0.08 | - | 0.069 | 69.94 | **0.02** | - | 0.048 |
| | **Uni-VQA (Ours)** | **72.98** | **0.07** | 69.86 | 0.440 | **74.95** | 0.06 | 64.89 | 0.391 |
| ViLBERT | Standard VQA | 66.98 | 0.19 | - | 0.009 | 69.23 | 0.20 | - | 0.004 |
| | VectorScale | 66.87 | 0.14 | - | 0.011 | 69.04 | 0.17 | - | 0.007 |
| | **Calibrated (Ours)** | 66.90 | **0.05** | - | 0.027 | 70.59 | **0.02** | - | 0.012 |
| | **Uni-VQA (Ours)** | **71.65** | 0.07 | 79.06 | 0.447 | **75.63** | 0.05 | 67.19 | 0.370 |
| VisualBERT | Standard VQA | 64.92 | 0.14 | - | 0.009 | 65.28 | 0.19 | - | 0.003 |
| | VectorScale | 64.83 | 0.14 | - | 0.010 | 64.40 | 0.18 | - | 0.003 |
| | **Calibrated (Ours)** | 65.26 | **0.03** | - | 0.027 | 67.38 | **0.01** | - | 0.009 |
| | **Uni-VQA (Ours)** | **70.95** | 0.08 | 77.87 | 0.440 | **74.34** | 0.06 | 73.46 | 0.399 |
| BEiT-3 | Standard VQA | 73.19 | 0.14 | - | 0.009 | 72.29 | 0.18 | - | 0.009 |
| | VectorScale | 73.62 | 0.14 | - | 0.009 | 72.16 | 0.16 | - | 0.009 |
| | **Calibrated (Ours)** | 73.25 | **0.04** | - | 0.027 | 71.94 | **0.02** | - | 0.027 |
| | **Uni-VQA (Ours)** | **74.33** | 0.07 | 35.91 | 0.217 | **76.01** | 0.02 | 57.82 | 0.333 |

in the high-confidence bins of the calibrated `TS-VQA` model—are confidently answered without further delegation to the LLM, provided their confidence surpasses the dynamic threshold. Consequently, the hybrid model achieves higher accuracy with reduced reliance on the LLM, underscoring the importance of calibration enhancement and the `Uni-VQA` approach.

Table 2: LLM-Delegation percentage comparison of proposed `Uni-VQA` against the LLM-VQA and LLM-VectorScale (threshold) baselines, to match the maximum accuracy achieved by baselines on VQA-v2.

| Model | ACC | LLM-Deleg (%)↓ | | ACC | LLM-Deleg (%)↓ | |
|---|---|---|---|---|---|---|
| | | LLM-VQA (thresh.) | Uni-VQA | | LLM-VectorScale (thresh.) | Uni-VQA |
| **Pythia** | 70.07 | 64.38 | **50.06** (-14.32%) | 70.07 | 66.11 | **50.06** (-16.05%) |
| **CLIP-ViL** | 71.51 | 35.5 | **24.4** (-11.1%) | 71.6 | 40.56 | **27.56** (-13.0%) |
| **ViLBERT** | 70.25 | 51.03 | **41.06** (-9.97%) | 70.42 | 60.86 | **41.06** (-19.8%) |
| **VisualBERT** | 69.75 | 64.01 | **47.51** (-16.5%) | 69.88 | 66.79 | **49.18** (-17.61%) |
| **BEiT-3** | 73.71 | 10.16 | **9.06** (-1.1%) | 73.62 | 26.23 | **6.71** (-19.52%) |

Our experiments reveal a trade-off between accuracy and the delegation percentage to the LLM. Adjusting the delegation threshold allows control over how often the LLM is used. Lowering the threshold reduces the reliance on the LLM and computational costs, but results in smaller gains in accuracy. This flexibility enables adaptation based on resource constraints and performance requirements, making the hybrid approach versatile for practical applications.

## 5.2 ABLATION STUDIES

We analyze LLM/VLLM inference costs of our `Uni-VQA` in terms of the delegation percentage versus model accuracy. Additional ablation studies are provided in Appendix G.

**LLM Inference Cost Analysis.** We study the effectiveness of our technique in terms of LLM inference cost. Table 3 shows the fraction of samples delegated by our `Uni-VQA` model in order to obtain the same accuracy (i.e., 69.09%) as that of the LLM-Only model where 100% samples have been answered by the LLM. As shown, with far fewer delegated samples to the LLM, we achieve a competitive accuracy. For example, using ViLBERT VQA backbone, our `Uni-VQA` achieves an accuracy of 69.09% (matching LLM-only accuracy) with only 19.4% delegation to the LLM. This means, for most of the samples we can leverage the cheaper `TS-VQA` model whereas, we can delegate only limited amount of low-confident samples to the LLM for the correction. Hence, we can maintain high predictive accuracy while being more efficient on LLM inference cost by significantly reducing the LLM reliance.

While our `Uni-VQA` shows significant accuracy gains across all backbones in VQA-v2, the improvement in BEiT-3 model is limited. This can be attributed to the fact that the BEiT-3 model already demonstrates a strong performance compared to the LLM. In such cases where the `TS-VQA` is already highly competent, accuracy gain from LLM delegation is naturally smaller.

Table 2 shows the percentage of delegated questions of our technique baselined against the two competitive baselines: LLM-VQA and LLM-VectorScale (threshold). As shown, to achieve the given accuracy, proposed `Uni-VQA` delegates significantly lower fraction of questions to LLM and thereby being computationally more efficient.

Table 3: Delegation percentage for `Uni-VQA` models to match LLM-Only accuracy.

| | Target ACC | LLM-Deleg (%) | | |
|---|---|---|---|---|
| | | Pythia | ViLBERT | VisualBERT |
| VQA-v2 | 69.09 | 38.81 | 19.4 | 40.39 |
| COCOQA | 72.03 | 17.14 | 8.06 | 12.13 |

Table 4: LLaVA Delegation across models, to match the LLaVA accuracy (78.35%).

| Model | LLM-Deleg (%) | | |
|---|---|---|---|
| | LLM-VQA | LLM-VectorScale | Uni-VQA (Ours) |
| CLIP-ViL | 80.3 | 73.8 | **65.4** |
| ViLBERT | 94.0 | 87.4 | **82.7** |
| VisualBERT | 93.2 | 89.9 | **84.6** |
| Pythia | 89.7 | 85.9 | **84.3** |

**VLLM Inference Cost Analysis** We also analyze the effectiveness of our `Uni-VQA` approach in reducing inference costs by using LLaVA, a large-scale vision language model (VLLM), as the LLM-based model. Our main objective is to demonstrate that `Uni-VQA` significantly lowers the reliance on LLaVA while maintaining comparable accuracy levels. Table 4 presents the LLM-delegation percentages for different VQA architectures, between our `Uni-VQA`, LLM-VQA and LLM-VectorScale (threshold), in order to achieve the same accuracy as the LLaVA-only setup, indicating a substantial reduction in delegation when using `Uni-VQA`. For instance, with CLIP-ViL as `TS-VQA` model, `Uni-VQA` achieves the same accuracy as LLaVA-only (78.35%) while requiring approximately $15\%$ and $8\%$ less delegation compared to LLM-VQA and LLM-VectorScale, respectively.

By leveraging the calibrated confidences of our `Calibrated TS-VQA` models, `Uni-VQA` effectively routes a fraction of questions to LLaVA only when necessary, avoiding redundant heavy computation on questions that can be reliably answered by the `TS-VQA`. Consequently, `Uni-VQA` not only reduces inference latency but also lowers the overall computational cost, making it a cost-effective alternative to relying fully on large models.

**Remark.** As Tables 2 and 4 indicate, we observe that the reduction in LLM delegation is more pronounced for models with well-calibrated confidence scores. This further emphasizes the role of calibration of `TS-VQA` models in enabling effective knowledge-exchange and uncertainty-aware integration between the `TS-VQA` and LLM.

## 5.3 Sensitivity & Robustness Analysis

We conduct a cross-model hyperparameter transfer analysis, where we applied hyperparameters $\{l, u, K(c_i)\}$ tuned on each model to other models, and measuring the impact on their performance, to analyze generalizability of our hyperparameter selections. Table 9 (in Appendix G.7) shows that the maximum accuracy drop never exceeds $1.24\%$ on COCO-QA, confirming that our proposed framework is not sensitive to careful tuning of the hyperparameters. Our analysis provides compelling evidence that careful threshold tuning is unnecessary, and thresholds show remarkable generalizability.

## 5.4 Discussion

Reducing reliance on computationally intensive models is crucial in ensuring scalable and environmentally sustainable AI applications, as studies have highlighted significant energy consumption and carbon emissions of large-scale language models (Strubell et al., 2020; Patterson et al., 2021). Our work addresses these concerns by minimizing frequent delegation to high-cost models through strategic integration. Unlike pruning (Zhu et al., 2024; Wan et al., 2023; Fu et al., 2024) and quantization (Zhao et al., 2024; Lin et al., 2024) techniques that reduce model size, our `Uni-VQA` approach improves inference efficiency by dynamically determining when LLM delegation is necessary based on calibrated `TS-VQA` confidence scores. This complementary approach can be combined with existing model efficiency techniques to further reduce computational costs while maintaining accuracy.

## 6 Conclusion

In this paper, we introduce an uncertainty-aware LLM integrated VQA model, referred to as `Uni-VQA`, which facilitates knowledge exchange between the LLM and a calibrated `TS-VQA` model based on reliable confidence scores. It cost-effectively improves VQA accuracy and inference speed. Our framework leverages well-calibrated confidence scores to guide the interaction between the LLM and `TS-VQA`. We conducted extensive experiments across multiple datasets, which demonstrate the effectiveness of `Uni-VQA` in terms of accuracy, computational efficiency, and reliability.

## ACKNOWLEDGMENTS

This research was sponsored by the Army Research Office and was accomplished under Grant Number W911NF-24-1-0385. The views and conclusions contained in this document are those of the authors and should not be interpreted as representing the official policies, either expressed or implied, of the Army Research Office or the U.S. Government. The U.S. Government is authorized to reproduce and distribute reprints for Government purposes notwithstanding any copyright notation herein.

## REPRODUCIBILITY STATEMENT

We provide: (1) Complete implementation details in Appendix F.3 with all hyperparameters in Table 18 (Appendix G.13); (2) Reference to public baseline implementations via MMF (Singh et al., 2020) and UniLM (Wang et al., 2023) repositories; (3) Standard public datasets (VQA-v2, COCO-QA) with preprocessing documented in Appendix F.1; (4) LLM prompting methodology in Appendix F.2; (5) Threshold learning procedure in Section 3.3 and F.4; and (6) code repository provided in supplementary materials, with public release at publication. Computational environment details are in Appendix G.12 (Tables 16-17).

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

# Supplementary Material

In this Appendix, we first present Table 5, summarizing the major notations used in our paper in Section A. Next, we provide the important concepts required for the Methodology in Section B. In Section C, we present the detailed methodology for training our diverse ensemble approach for VQA calibration. In Section D we provide the detailed mathematical proofs for our theoretical contributions. In Section E we provide additional related works. We provide additional experimental details along with the results in Section F, a detailed Ablation study in Section G, and additional qualitative analysis in Section H. Finally, we provide the broader impact statement and limitations associated with our work in Sections I and J, respectively.

## A  SUMMARY OF NOTATIONS

Table 5 summarizes the major notations used in our paper.

Table 5: Summary of Notation

| Symbol Group | Notation | Description |
|---|---|---|
| DATASET | $\mathcal{A}$ | Answer set |
| | $\mathcal{V}$ | Image set |
| | $\mathcal{Q}$ | Question set |
| | $\mathcal{V} \times \mathcal{Q}$ | Input set |
| | $\mathbf{x}_n \equiv (\mathbf{v}_n, \mathbf{q}_n)$ | Input image-question pair |
| | $C$ | Total number of classes |
| DRO LOSS | $D_f$ | $f$-divergence |
| | $l(\mathbf{x}, \Theta)$ | Per-sample loss |
| | $\lambda$ | DRO loss parameter |
| | $p_y^n$ | Output probability for the $n$-th sample associated with class $y$ |
| HYBRID VQA | $K$ | Number of answer candidates from `TS-VQA` |
| | $c_i$ | Confidence of predicted answer given input $\mathbf{x}_i$ |
| | $K(c_i)$ | Dynamically chosen candidate count based on output confidence |

## B  PRELIMINARIES

In this section, we provide the key concepts that are required to understand our approach.

**VQA Accuracy:**  In the Visual Question Answering (VQA) task, each question is associated with multiple ground-truth answers provided by human annotators. Let $\mathbf{a}$ denote the set of ground-truth answers for a given question, and let $\hat{a}$ represent the answer predicted by a VQA model. The VQA accuracy metric is defined as follows:

$$Acc(\hat{a}, \mathbf{a}) = \min\left(1, \frac{\#\text{ answers in } \mathbf{a} \text{ matching } \hat{a}}{3}\right).$$

**Expected Calibration Error (ECE):**  (Naeini et al., 2015) is a metric commonly used to assess the calibration error between the estimated confidences and the actual accuracies. ECE is calculated by dividing the $N$ predictions into $M$ equal bins according to their confidence scores. Within each bin $B_m$, the average accuracy and confidence are denoted by $\text{acc}(B_m)$ and $\text{conf}(B_m)$. Then, ECE is calculated as (Guo et al., 2017):

$$\text{ECE} = \sum_{m=1}^{M} \frac{|B_m|}{N} \left|\text{acc}(B_m) - \text{conf}(B_m)\right|,$$

where $|B_m|$ is the number of samples in the $m$-th bin. In the context of VQA, where there is more than a single ground-truth answer, ECE is measured with respect to the most frequent answer in the ground-truth annotations.

**Adaptive Calibration Error (ACE):** (Nixon et al., 2019) is an alternative metric to measure calibration, which measures the difference between the confidences and accuracies across all classes, with adaptive binning rather than static and fixed-width binning as in ECE. In contrast, ACE divides the interval $[0, 1]$ into bins with equal number of samples. ACE is defined as:

$$\text{ACE} = \frac{1}{M|A|} \sum_{y=1}^{|A|} \sum_{m=1}^{M} \big| \text{acc}(m, y) \ - \ \text{conf}(m, y) \big|$$

where $m \in \{1, \ldots, M\}$ indexes the bins, $y \in \{1, \ldots, |A|\}$ indexes the classes, $|A|$ is the total number of classes, $\text{acc}(m, y)$ is the empirical accuracy in bin $m$ for class $y$, and $\text{conf}(m, y)$ is the mean predicted confidence for the same bin and class.

**Brier Score:** (Brier, 1950) measures the squared error difference between the confidences and actual accuracies, without binning, and is defined as:

$$\text{Brier} = \frac{1}{N} \sum_{n=1}^{N} (p_i - y_i)^2,$$

where $p_i$ and $y_i$ represent the confidence, and the prediction accuracy for the $i$th sample.

**Negative Log Likelihood (NLL):** (Friedman et al., 2001) is also known as cross-entropy loss, and is defined as:

$$\text{NLL} \ = \ -\frac{1}{N} \sum_{i=1}^{N} \log p(y_i | x_i),$$

where $p(y_i | x_i)$ are the predicted probabilities of the ground-truth to the true targets for the $i$th input.

## C  DETAILED METHODOLOGY: DIVERSE ENSEMBLE FOR VQA CALIBRATION

Our diverse ensemble approach builds upon Distributionally Robust Optimization (DRO) (Duchi & Namkoong, 2019), which seeks to minimize the worst-case expected loss over an uncertainty set of distributions. The standard DRO formulation is:

$$\mathcal{L}_{DRO}(\Theta) = \max_{\mathbf{w} \in \mathcal{W}} \sum_{n=1}^{N} \mathbf{w}_n l(\mathbf{x}_n, \Theta) \tag{5}$$

where $\mathcal{W}$ is the uncertainty set defined as:

$$\mathcal{W} := \left\{ \mathbf{w} \in \mathbb{R}^N : \mathbf{w}^\top \mathbf{1} = 1, \mathbf{w} \geq 0, D_f\left(\mathbf{w} \| \frac{\mathbf{1}}{N}\right) \leq \frac{\lambda}{N} \right\} \tag{6}$$

Here, $D_f(\mathbf{w} \| \frac{1}{N})$ measures the f-divergence between the weight distribution $\mathbf{w}$ and the uniform distribution $\frac{1}{N}$, and $\lambda$ controls the size of the uncertainty set.

To make the optimization tractable, we employ the regularized version with KL-divergence as the f-divergence measure. The closed-form solution for the optimal weights becomes:

$$w_n^*(\lambda) = \frac{\exp(l(\mathbf{x}_n, \Theta)/\lambda)}{\sum_{j=1}^{N} \exp(l(\mathbf{x}_j, \Theta)/\lambda)} \tag{7}$$

This softmax-like weighting scheme has intuitive properties: (1) **High Loss Emphasis**: Samples with higher losses $l(\mathbf{x}_n, \Theta)$ receive exponentially higher weights, (2) **Temperature Control**: The

parameter $\lambda$ acts as a temperature parameter controlling the concentration of weights, (3) **Normalization**: The weights sum to 1, maintaining a valid probability distribution.

**Effect of Hyperparameter $\lambda$ on Model Specialization** The hyperparameter $\lambda$ fundamentally determines the focus of each ensemble member:

*Case 1: Small $\lambda$ (Hard Sample Expert)*
When $\lambda \to 0$, the weight computation becomes:

$$\lim_{\lambda \to 0} w_n^*(\lambda) = \begin{cases} \frac{1}{|\mathcal{H}|} & \text{if } n \in \mathcal{H} = \arg\max_j l(\mathbf{x}_j, \Theta) \\ 0 & \text{otherwise} \end{cases} \tag{8}$$

where $\mathcal{H}$ is the set of hardest samples. This creates a model that focuses exclusively on the most challenging examples.

*Case 2: Large $\lambda$ (General Pattern Expert)*
When $\lambda \to \infty$, the weights approach uniform distribution:

$$\lim_{\lambda \to \infty} w_n^*(\lambda) = \frac{1}{N}, \quad \forall n \tag{9}$$

This is equivalent to standard Empirical Risk Minimization (ERM), producing a model that captures general data patterns, shows higher confidence on typical samples, achieves good average performance.

*Case 3: Moderate $\lambda$ (Balanced Expert)*
Intermediate values of $\lambda$ create models that balance between hard and easy samples.

## D    MATHEMATICAL PROOFS

In this section, we provide the mathematical proof for Lemma 4.1 and Theorem 4.2.

### D.1    PROOF OF LEMMA 4.1

The DRO loss (Sapkota & Yu, 2023) can be written as the following:

$$\mathcal{L}_{DRO}(\theta) = -\sum_{y=1}^{|A|} \frac{\exp\left(-\frac{\log(\hat{p}_y)}{\lambda}\right)}{C^{DRO}} \log(\hat{p}_y) \tag{10}$$

Where $\hat{p}_y$ is the predictive distribution, $\lambda$ is the DRO regularizer coefficient, $C^{DRO}$ is the normalization constant and $|A|$ being total number of classes. We can write the following inequality

$$\mathcal{L}_{DRO}(\theta) \geq -\frac{1}{C^{DRO}} \sum_{y=1}^{|A|} (1 - \lambda \hat{p}_y) q_y \log \hat{p}_y \tag{11}$$

Where $q_y$ is the ground truth probability assigned to $y^{th}$ class with $q_y = 1$ if $y = a(answer)$ and $q_y = 0$ otherwise.
$\forall y, \log(\hat{p}_y) \leq 0$ we can write the following:

$$\mathcal{L}_{DRO}(\theta) \geq -\frac{1}{C^{DRO}} \left[ \sum_{y=1}^{|A|} q_y \log(\hat{p}_y) - \lambda \left| \sum_{y=1}^{|A|} q_y \hat{p}_y \log(\hat{p}_y) \right| \right]$$

$$\geq -\frac{1}{C^{DRO}} \left[ \sum_{y=1}^{|A|} q_y \log(\hat{p}_y) - \lambda \max_j q_j \sum_{y=1}^{|A|} |\hat{p}_y \log(\hat{p}_y)| \right]$$

By Holder inequality $||fg||_1 \leq ||f||_\infty ||g||_1$ we can further rewrite the above equation as follows

$$\mathcal{L}_{DRO}(\theta) \geq -\frac{1}{C^{DRO}} \left[ \sum_{y=1}^{|A|} q_y \log(\hat{p}_y) - \lambda \sum_{y=1}^{|A|} \hat{p}_y \log(\hat{p}_y) \right]$$

$$= \frac{1}{C^{DRO}}[\mathcal{L}_{CE}(\theta) - \lambda \mathcal{H}[\hat{p}]]$$

Let $\lambda_1, ... \lambda_E$ be the DRO specific parameters for the $E$ ensemble models and $C_e^{DRO}$ be the respective normalization constant then we can write the following:

$$\sum_{e=1}^{|E|} \mathcal{L}^{DRO}(\theta) \geq \sum_{e=1}^{|E|} \frac{1}{C_{DRO}^e} [\mathcal{L}_{CE}^e(\theta) - \lambda_e \mathcal{H}^e[\hat{p}]] \tag{12}$$

Consider, $C \in \min_{e \in |E|}\{C_{DRO}^e\}$ then we have the following

$$L_{DE}(\theta) \geq \frac{1}{C} \sum_{e=1}^{|E|} [\mathcal{L}_{CE}^e(\theta) - \lambda_e \mathcal{H}^e[\hat{p}]] \tag{13}$$

This proves the Lemma.

*Steps from Eq. 10 to 11:*

We can rewrite the following:

$$\exp\left(-\frac{\log(\hat{p}_y)}{\lambda}\right) = (\exp(\log(\hat{p}_y)))^{-\frac{1}{\lambda}} = \hat{p}_y^{-\frac{1}{\lambda}}$$

**Case 1: if $\hat{p}_y \lambda \geq 1$:** In this case $(1 - \lambda \hat{p}_y) \leq 0$ and $\hat{p}_y^{-\frac{1}{\lambda}} \geq 0$ and therefore $\hat{p}_y^{-\frac{1}{\lambda}} \geq (1 - \lambda \hat{p}_y)$

**Case 2: if $\hat{p}_y \lambda < 1$:** In this case as $\hat{p}_y < 1$, and therefore $\hat{p}_y^{-\frac{1}{\lambda}} > 1$ whereas $(1 - \lambda \hat{p}_y) < 1$ and therefore $\hat{p}_y^{-\frac{1}{\lambda}} \geq (1 - \lambda \hat{p}_y)$ As in both cases, $\hat{p}_y^{-\frac{1}{\lambda}} \geq (1 - \lambda \hat{p}_y)$ and therefore Eq. 11 leads from Eq. 10

### D.2 PROOF OF THEOREM 4.2

Based on Lemma 4.1, minimizing our DE loss ensures increase in the entropy.

We first establish the relationship between entropy and confidence through general bounds that hold for any probability distribution, then show that increasing entropy lowers the admissible confidence bounds.

ENTROPY-CONFIDENCE BOUNDS

We first prove that entropy constrains confidence to lie within a bounded interval.

**Lemma D.1 (Entropy-Confidence Bounds)** *Let $p$ be a probability distribution over the $A$ classes and let $c = \max_i p_i$, denote the confidence. Define the binary entropy $h(c) = -c \log c - (1 - c) \log(1 - c)$. Then the Shannon entropy $H(p)$ satisfies:*

$$-\log c \leq H(p) \leq h(c) + (1 - c) \log(A - 1) \tag{14}$$

**Lower bound:** The lower bound follows from the relationship between Shannon entropy and min-entropy (Rényi entropy of order infinity). By definition:

$$H(p) = -\sum_i p_i \log p_i \geq -\sum_i p_i \log c = -\log c$$

where the inequality holds because $p_i \leq c$ for all $i$, which implies $-\log p_i \geq -\log c$.

**Upper bound:** Without loss of generality, assume the maximum probability is $p_1 = c$. Let $r = (r_2, \ldots, r_A)$ be the distribution over the remaining $A-1$ classes, i.e., $p_i = (1-c)r_i$ for $i = 2, \ldots, A$ with $\sum_{i=2}^{A} r_i = 1$. Then:

$$
\begin{aligned}
H(p) &= -c \log c - \sum_{i=2}^{A} p_i \log p_i \\
&= -c \log c - \sum_{i=2}^{A} (1-c)r_i \left( \log(1-c) + \log r_i \right) \\
&= -c \log c - (1-c) \log(1-c) - (1-c) \sum_{i=2}^{A} r_i \log r_i \\
&= h(c) + (1-c)H(r)
\end{aligned}
$$

Since $r$ is a distribution over $A-1$ classes, we have $H(r) \leq \log(A-1)$ (with equality when $r$ is uniform). Thus:

$$
H(p) \leq h(c) + (1-c) \log(A-1)
$$

CONFIDENCE UPPER BOUND AS A FUNCTION OF ENTROPY

Define $F_A(c) = h(c) + (1-c) \log(A-1)$. We show that this function is strictly decreasing for $c > \frac{1}{A}$:

$$
\begin{aligned}
F_A'(c) &= \frac{d}{dc} \left[ -c \log c - (1-c) \log(1-c) + (1-c) \log(A-1) \right] \\
&= -\log c - 1 + \log(1-c) + 1 - \log(A-1) \\
&= \log \frac{1-c}{c} - \log(A-1) = \log \frac{1-c}{c(A-1)}
\end{aligned}
$$

For $c > \frac{1}{A}$, we have $c(A-1) > 1-c$, which implies $\frac{1-c}{c(A-1)} < 1$, and thus $F_A'(c) < 0$. Therefore, $F_A$ is strictly decreasing on $(\frac{1}{A}, 1]$ and hence invertible. Let $g_A$ denote its inverse.

From the entropy bounds in Lemma D.1, we obtain bounds on confidence as a function of entropy:

$$
e^{-H(p)} \leq c \leq g_A(H(p)) \tag{15}
$$

**Key Implication:** As entropy $H(p)$ increases, both the lower and upper bounds on confidence $c$ decrease. This means that *high confidence values become incompatible with high entropy*, regardless of how the non-maximum probabilities are distributed.

**Example (VQA-v2 with $A = 3129$):** At $H(p) = 1$, the upper bound gives $c \leq 0.91$; at $H(p) = 4$, we have $c \leq 0.59$. Thus, moderately increased entropies already exclude very high confidences.

COMPLETING THE PROOF OF THEOREM 4.2

Minimizing our DE loss ensures the increase in the entropy, which makes confidence $\hat{p}$ lower than that of the ERM loss. We can state this fact in expectation: $\mathbb{E}[\hat{p}_{DE}] \leq \mathbb{E}[\hat{p}_{ERM}]$.

Considering the equal accuracy assumption between ERM and DE, we can write the following:

$$
\mathbb{E}[P(\hat{y} \neq y)]_{DE} \approx \mathbb{E}[P(\hat{y} \neq y)]_{ERM} \tag{16}
$$

Now let's break this into the high $(> \tau)$ and low confidence region $(< \tau)$. We can write the following:

$$\mathbb{E}[P(\hat{y} \neq y)]_{DE}^{<\tau} + \mathbb{E}[P(\hat{y} \neq y)]_{DE}^{>\tau} \approx \mathbb{E}[P(\hat{y} \neq y)]_{ERM}^{<\tau} + \mathbb{E}[P(\hat{y} \neq y)]_{ERM}^{>\tau} \qquad (17)$$

Let us consider $N_{ERM,in}^{>\tau}$ be the number of incorrectly classified samples in the high-confidences region in ERM and $N_{DE,in}^{>\tau}$ be the samples in DE. We make an assumption that the number of confidently wrong samples are higher in ERM. This has been observed in our empirical evaluation (Figure 11) as well as found in the existing literature (e.g. see Fig. C.2 in Mukhoti et al. (2020)). Based on this expectation and invoking the fact that $\mathbb{E}[\hat{p}_{DE}] \leq \mathbb{E}[\hat{p}_{ERM}]$, the incorrect samples using DE will be pushed more toward the low confidence region. This will lead to the following

$$\mathbb{E}[P(\hat{y} \neq y)]_{DE}^{>\tau} \leq \mathbb{E}[P(\hat{y} \neq y)]_{ERM}^{>\tau} \qquad (18)$$

Above equation immediately leads to the following:

$$\mathbb{E}[P(\hat{y} \neq y)]_{DE}^{<\tau} \geq \mathbb{E}[P(\hat{y} \neq y)]_{ERM}^{<\tau} \qquad (19)$$

This proves our Theorem. Our empirical findings, shown in Figs. 11 and 12, support this, as they demonstrate that our calibrated model has more samples in the less confident region compared to the uncalibrated Standard VQA. Fig. 5 empirically validate that $N_{DE,in}^{<\tau} \geq N_{ERM,in}^{<\tau}$ hold. Additionally, Fig. 8 validate that $N_{DE}^{\tau} \geq N_{ERM}^{\tau}$.

**Empirical Support for the Relationship Between Entropy and Confidence:** We provide extensive empirical validation of our theoretical analysis: (a) *Per-sample analysis:* Among all samples with increased entropy under our method, **96.61%** also exhibit decreased confidence (see Fig. 9b), providing strong empirical support for the inverse relationship. (b) *Correlation analysis:* The Pearson correlation between entropy and confidence is $r = -0.9236$, and the Spearman rank correlation is $\rho = -0.9604$ over the validation set. Both statistics confirm a pronounced inverse relationship. (c) *Bound verification:* Fig. 9a plots empirical $(H(p(x)), \max p(x))$ pairs together with the theoretical lower bound $e^{-H}$ and upper bound $g_A(H)$, demonstrating that all predictions fall within the admissible region as predicted by Lemma D.1.

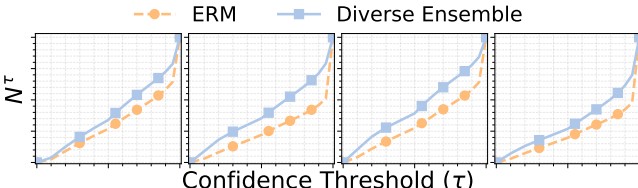

Figure 8: Empirical evidence illustrating $N_{DE}^{\tau} \geq N_{ERM}^{\tau}$, across four VQA architectures.

# E    ADDITIONAL RELATED WORK

**Model Cascades:** Our proposed framework relates to several research directions in the literature. While we discussed some related works in Section 2, our approach also shares common goals with model cascades (Wang et al., 2017; Warren & Dras, 2025; Jitkrittum et al., 2023; Enomoto & Eda, 2021; Rabanser et al., 2025), that aims at reducing the computational efficiency by strategically routing inputs through a cascade of models with progressively increasing capacity, complexity and computational costs, based on deferral mechanisms, hence enabling easy inputs to be handled by cheaper and simpler models, while complex inputs being progressively cascaded to the more complex models. In this section, we elaborate on the connections between our approach and existing approaches in model cascades, highlighting their key distinctions setting our work apart.

Model cascades are often used to improve inference efficiency by sequentially routing harder inputs to more sophisticated models, when earlier ones are uncertain, where a deferring mechanism determines whether to defer to a large model, or accept the current model's output. Common deferring mechanisms rely on confidence or uncertainty estimates from smaller model. Sharing the same goal, our method is different than existing method in the model cascades literature. Methods including IDK cascades (Wang et al., 2017), rely solely on raw confidence scores (typically the maximum softmax probabilities) without applying any explicit calibration. However, recent works (Jitkrittum et al., 2023; Enomoto & Eda, 2021; Rabanser et al., 2025) highlighted that uncalibrated confidences

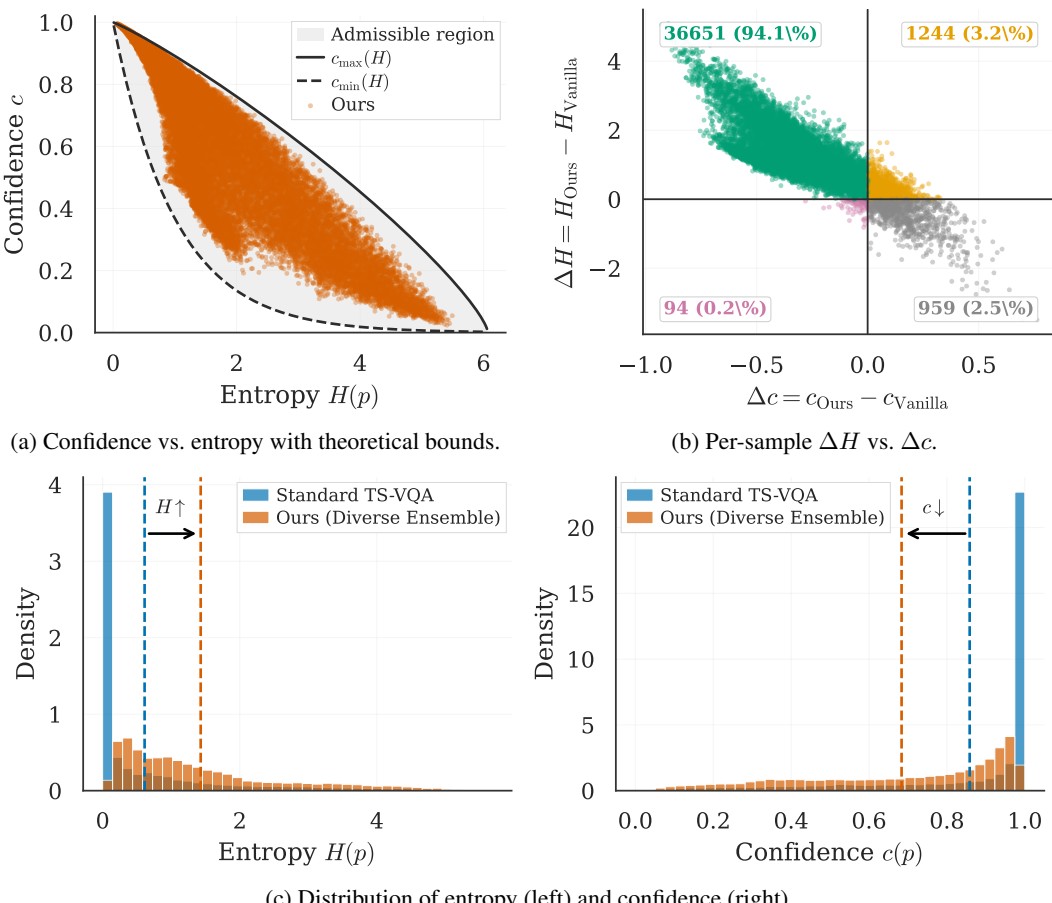

(a) Confidence vs. entropy with theoretical bounds.

(b) Per-sample $\Delta H$ vs. $\Delta c$.

(c) Distribution of entropy (left) and confidence (right).

Figure 9: Entropy-confidence analysis. (a) Empirical predictions within the admissible entropy-confidence region. (b) Quadrant analysis shows our method increases entropy while reducing confidence for the majority of samples. (c) Distribution shift: mean entropy increases from 0.61 to 1.43; mean confidence decreases from 0.86 to 0.68.

can lead to suboptimal deferral decisions, especially when the downstream model behaves as a specialist. In contrast, other methods proposed explicit confidence calibration techniques to improve deferral, such as learning-to-cascade (LtC) (Enomoto & Eda, 2021) and gatekeeper-based tuning (Rabanser et al., 2025), both of which improve routing decisions by improving the calibration of the smaller model's confidence estimates. Nevertheless, these works consider simple routing as a deferring mechanism, i.e. if the calibrated confidence of the smaller model falls below a threshold, the input is routed to a large model which predicts the final answer. In contrast, our approach introduces adaptive integration by enabling the simple model to adaptively share knowledge with the larger model through candidate answers, which, as supported by our experiments, lead to more informed reasoning and significant improvements in our overall cascade accuracy.

# F    ADDITIONAL EXPERIMENTAL DETAILS

In this section, we first provide a detailed description of the datasets, followed by an explanation of the LLM prompt construction and in-context example selection. Next, we provide the implementation details of our technique. After that we show the ECE plot of our technique along with other competitive baselines. Finally, we show the performance of the hybrid approaches where we integrate different baselines with LLMs.

## F.1 DATASET DESCRIPTION

We experiment on the VQA-v2 (Antol et al., 2015) and COCO-QA (Ren et al., 2015) datasets, which contain questions on the COCO image dataset (Lin et al., 2014). VQA-v2 dataset consists of $443,757$ questions in training split, and 10 ground-truth answers per each question. As the ground-truth answers of the test split of VQA-v2 are not publicly available, we use the validation and test splits as provided by (Whitehead et al., 2022), as evaluating the calibration error requires sample-level accuracies. The test split consists of 106k, and the validation split consists of 86k questions.

COCO-QA dataset contains $78,736$ training, $38,948$ testing questions generated from Microsoft COCO dataset (Lin et al., 2014), with a single ground-truth answer per question. In experiments, we randomly sample a validation split of size of 12000 from the training set.

## F.2 LLM-BASED INFERENCE FOR VQA

We describe the process of delegating question answering to LLMs when the predicted confidence score of the `TS-VQA` model falls below a predefined threshold $\tau$. We outline the in-context learning based paradigm for prompting the LLM, and the procedure for constructing effective prompts. For LLM-based prompting for VQA task, we follow prior works (Yu et al., 2023; Yang et al., 2022). To leverage the LLM, we use the few-shot in-context learning approach, which is an effective approach to adapt the LLM to a certain task, without the need for computationally intensive fine-tuning, by augmenting the prompt with input and output examples, enabling an efficient and training-free adaptation to the task.

### F.2.1 PROMPT CONSTRUCTION

Creating a structured input prompt for the LLM involves several components that help the LLM understand the question's context and generate accurate answers. The prompt is structured as shown in the template below, where underlined text represents template keywords, and the rest are place-holders for the data samples.

```
Context: c \n Question: q \n Answer: a
```

### F.2.2 CONTEXTUAL INFORMATION

To help the LLM model comprehend the visual content referenced in the question, we use off-the-shelf image captioning models to generate descriptive of the image in textual format. Similar to prior works (Guo et al., 2023), we leverage the PNP-VQA model (Tiong et al., 2022) for image captioning, which generates captions relevant to the question, ensuring that the LLM has relevant contextual information to answer the question.

### F.2.3 IN-CONTEXT EXAMPLES

In-context examples consist of an example prompt, along with the desired answers from the training data, formatted similarly to the test prompt. These examples help the LLM generate the answer by following the pattern established in the prompt. For each test sample, multiple in-context examples are selected based on their cosine similarity to the test image-question pairs. This involves extracting the image and text embeddings from the VQA data, using an off-the-shelf pretrained model. We specifically use BLIP-2 model (Li et al., 2023) for this purpose. The average cosine similarity between the embeddings of any two image-question pairs $(\mathbf{q}_i, \mathbf{v}_i)$ and $(\mathbf{q}_j, \mathbf{v}_j)$ in training and test splits is calculated. The top $N$ examples with the highest similarity are then chosen as in-context examples.

### F.2.4 ANSWER-CANDIDATES AUGMENTED PROMPTS

As demonstrated in Fig. 2d, the predictive performance of the LLM can be enhanced when the prompt is augmented with some answer candidates. Assume that given an input $\mathbf{x}_i$ to the task-specifc VQA model, $\hat{\mathcal{A}}_K = \{\hat{a}_1, \cdots, \hat{a}_K\}$ are the $K$ candidate answers corresponding to the $K$ answers with highest probabilities in descending order, and $\mathcal{C}_K = \{\hat{p}_1, \cdots, \hat{p}_K\}$ are the corresponding probabilities. Given the set of $K$ answer candidates, we augment the prompt and present

a set of answer candidates as additional context to the question. The answer candidate augmented prompt is constructed as bellow:

```
Context: c \n Question: q \n Candidates: C \n
                  Answer: a
```

## F.3 IMPLEMENTATION DETAILS

In this section, we provide additional implementation and experimental details of our proposed method and experiments. We conducted our experiments using PyTorch. Our experiments utilize publicly available implementations across all models. For all VQA architectures except for BEiT-3, we use their implementations as provided by the *MMF* (Singh et al., 2020) repository[2]. For BEiT-3, we use its official implementation from Microsoft's UniLM project (Wang et al., 2023)[3]. To train the standard VQA models, the training hyperparameters of the networks given in *MMF* and *UniLM* repositories are used. For training `Calibrated` VQA models, we use the same training hyperparameters as the standard VQAs. We adopt the VectorScale implementation from Whitehead et al. (2022)[4].

We trained BEiT-3 `TS-VQA` on a single A100-40 GB GPU, and the rest of the `TS-VQA` models on a single NVIDIA RTX A6000-48 GB GPU. Furthermore, the latencies and carbon emissions in Fig. 1 and Table 1 are reported based on the models running on a single A100-40 GB GPU. For LLM inference with Mistral-7B, we run the model on a single A100-40 GB GPU.

**VQA by the LLM Model**   Following (Yu et al., 2023; Yang et al., 2022) we provide 9 captions as context for the question, and use PNP-VQA (Tiong et al., 2022) for generating question-related captions, as the context about images in the prompt. For each test instance, 10 in-context examples from the training data are selected based on the average of their image and question embedding cosine similarities, and included in the prompt. Specifically, BLIP-2 model is used to extract the image and question embeddings, used for in-context example selection. The LLM is queried 5 times to ensemble the answers as the final answer to the question. For answer-candidates-augmented VQA with LLM, we restrict to using 10, 5, 2, and 1 answer candidates. The LLM-based inferences are conducted once.

## F.4 CONFIDENCE THRESHOLD DETERMINATION AND DYNAMIC CANDIDATE SELECTION

In this section, we provide a detailed explanation of how the confidence thresholds $l$ and $u$, as well as the dynamic answer candidate selection function $K(c_i)$, are determined using a held-out validation set. Our approach is fully data-driven and optimizes for accuracy while structurally ensuring efficiency through selective LLM delegation.

**Optimization Objective and Process:** The threshold selection process is designed to maximize VQA accuracy on the validation set within each confidence region, while efficiency gains emerge naturally from the threshold structure itself. Specifically: (a) **(Primary objective)** We maximize validation accuracy within each confidence bin, and (b) **(Efficiency mechanism)** by setting threshold $u$ based on where `TS-VQA` achieves best accuracy (or alternatively dynamically based on computational budget), we automatically avoid unnecessary LLM invocations in high-confidence regions.

The complete process consists of two main steps:

**Step 1: Per-confidence-Bin Policy Selection.** We first partition the confidence range $[0, 1]$ into $B$ equal-width bins (we use $B = 10$ in our experiments). For each bin $b$, we evaluate multiple interaction modes on the validation set. *(1) `TS-VQA` only*, *(2) LLM-only*, without any answer candidates from `TS-VQA`, *(3) LLM with top-K candidates* ($K \in \{1, 2, 5, 7, 10\}$) from `TS-VQA`.

---

[2]https://github.com/facebookresearch/mmf
[3]`https://github.com/microsoft/unilm/tree/master/beit3`
[4]`https://github.com/facebookresearch/reliable_vqa`

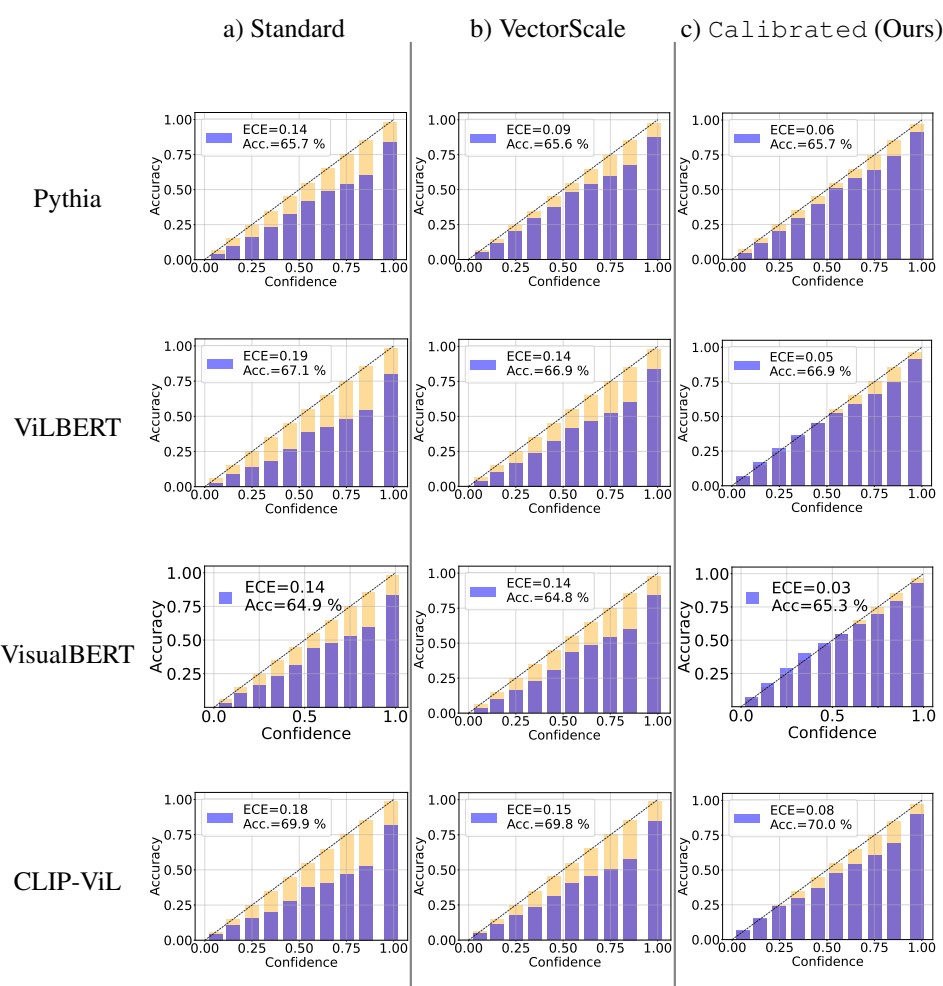

Figure 10: Effectiveness of DE-based VQA in improving the calibration of all VQA architectures. a) Standard, b) VectorScale-calibrated, and c) DE-based VQA models.

For each bin $b$ and each mode $m$, we compute the VQA accuracy on validation samples whose TS-VQA confidence falls within bin $b$. We then select the mode that maximizes accuracy for that bin:

$$m_b^* = \arg \max_{m \in \mathcal{M}} \mathrm{Acc}(m, b) \tag{20}$$

where $\mathcal{M} = \{\text{TS-VQA}, \text{LLM}_{\text{top-0}}, \text{LLM}_{\text{top-1}}, \ldots\}$. This creates a discrete mapping from confidence bins to optimal interaction modes, ensuring that we only invoke the LLM (and only with a specific number of candidates) in regions where it demonstrably improves accuracy on held-out data.

**Step 2: Deriving Thresholds and Smooth $K(c)$ Function**. From the bin-wise optimal policies determined in *Step 1*, we derive the continuous thresholds and candidate selection function:

- Lower threshold $l$: We define $l$ as the upper boundary of the highest confidence bin where "LLM-only (Top-0)" achieves the best accuracy. This identifies the region where TS-VQA has minimal domain knowledge and the LLM should answer without potentially misleading candidates.

- Upper threshold $u$: We define $u$ as the lower boundary of the lowest confidence bin where "TS-VQA only" achieves the best accuracy. This identifies the region where TS-VQA is sufficiently reliable to answer without LLM consultation.

- Dynamic candidate function $K(c_i)$: For the intermediate region $[l, u]$, we fit a smooth, monotonically decreasing function that maps confidence scores to the optimal number of

Table 6: Delegation percentage for hybrid models to match LLM-Only accuracy (72.03%) on COCO-QA dataset.

| LLM-Only | Uni-VQA (Ours) | | | |
|---|---|---|---|---|
| | Pythia | ViLBERT | VisualBERT | CLIP-ViL |
| 100 | 18.14 (-81.86%) | 8.06 (-91.94%) | 25.56 (-74.44%) | 12.13 (-87.87%) |

answer candidates. We use the exponential form presented in eq. (2) of the main paper, where the parameters $M$ and $W$ are learned by fitting to the per-bin optimal $K$ values determined in Step 1, subject to the constraints that $K(l) \approx M$ (maximum candidates at the lower threshold) and $K(c) \to 1$ as $c \to u$ (minimum candidates near the upper threshold).

**Rationale and Trade-offs:** This data-driven approach offers several advantages: (1) Accuracy-optimized: By selecting the best-performing mode for each confidence region on validation data, we ensure that delegation decisions are evidence-based rather than heuristic, (2) Efficiency through structure: The threshold $u$ naturally limits LLM usage to cases where it provides value, as high-confidence samples are handled by the calibrated `TS-VQA`, (3) Adaptive candidate selection: The smooth function $K(c_i)$ avoids abrupt changes in the number of candidates provided, ensuring that the LLM receives appropriate amounts of specialized knowledge based on `TS-VQA` uncertainty.

## G ABLATION STUDY

### G.1 ADDITIONAL EXPERIMENTS ON COCO-QA DATASET

Table 6 presents additional resuls on the COCO-QA dataset, illustrating the extent of LLM-delegation necessary for the hybrid models to attain equivalent accuracy as the LLM (Mistral-7B) on the COCO-QA dataset for each `TS-VQA` model. BEiT-3 is omitted since the BEiT-3 `TS-VQA` already surpasses the accuracy of Mistral-7B model on COCO-QA.

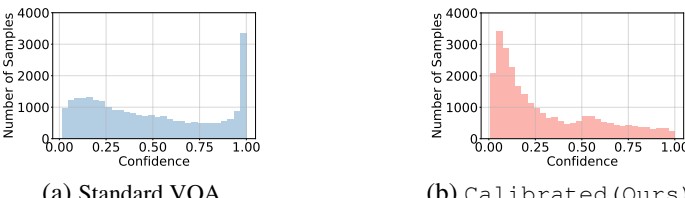

(a) Standard VQA          (b) `Calibrated(Ours)`

Figure 11: Confidence distribution of incorrect answers in a) Standar, and b) our `Calibrated` VQA.

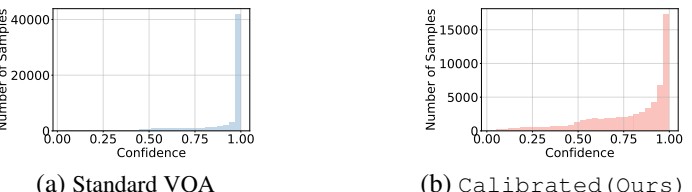

(a) Standard VQA          (b) `Calibrated(Ours)`

Figure 12: Confidence distribution of *correct* answers in a) Standard, and b) our `Calibrated` VQA.

### G.2 EFFECTIVENESS OF DIVERSE ENSEMBLE (DE)-BASED VQA CALIBRATION TOWARDS CALIBRATED VQA

In this subsection, we analyze the effectiveness of the DE-based framework in improving the calibration of `TS-VQA` models compared to standard and VectorScale-based VQA models. We present reliability diagrams for all four VQA architectures to illustrate the differences in calibration performances. Fig. 10 clearly shows that standard VQA models are overconfident and poorly calibrated, while VectorScale-based VQA models exhibit only a slight improvement in calibration, still suffering from overconfidence. In contrast, our proposed DE-based VQA significantly reduces the Expected Calibration Error (ECE) and overconfidence compared to the baselines, resulting in substantially improved reliability. These findings underscore the importance of employing effective

Table 7: Performance comparison of `Uni-VQA` with baseline `TS-VQA` models and LLM across four architectures, evaluated using multiple calibration metrics including ECE, ACE, Brier score, and NLL.

| | Model | VQA-v2 | | | | | COCO-QA | | | | |
|---|---|---|---|---|---|---|---|---|---|---|---|
| | | ACC↑ | ECE↓ | ACE↓ | Brier↓ | NLL↓ | ACC↑ | ECE↓ | ACE↓ | Brier↓ | NLL↓ |
| | LLM-only (Mistral-7B) | 69.09 | 0.31 | 0.34 | 0.30 | 0.91 | 72.03 | 0.27 | 0.48 | 0.27 | 0.94 |
| **Pythia** | Standard VQA | 65.66 | 0.14 | 0.13 | 0.19 | 0.67 | 68.62 | 0.16 | 0.16 | 0.19 | 0.80 |
| | VectorScale | 65.59 | 0.09 | 0.08 | 0.18 | 0.56 | 68.88 | 0.10 | 0.08 | 0.17 | 0.61 |
| | **Calibrated (Ours)** | 66.15 | **0.06** | **0.05** | **0.17** | **0.53** | 68.64 | **0.02** | **0.02** | **0.15** | **0.47** |
| | **Uni-VQA (Ours)** | **71.00** | 0.05 | **0.05** | 0.17 | 0.53 | **74.78** | 0.06 | 0.13 | 0.17 | 0.51 |
| **CLIP-ViL** | Standard VQA | 69.95 | 0.18 | 0.17 | 0.20 | 0.87 | 70.38 | 0.15 | 0.14 | 0.18 | 0.71 |
| | VectorScale | 69.81 | 0.15 | 0.14 | 0.19 | 0.67 | 70.41 | 0.11 | 0.09 | 0.17 | 0.58 |
| | **Calibrated (Ours)** | 70.05 | 0.08 | **0.07** | **0.16** | **0.52** | 69.94 | **0.02** | **0.03** | **0.15** | **0.47** |
| | **Uni-VQA (Ours)** | **72.98** | **0.07** | 0.07 | 0.17 | 0.53 | **74.95** | 0.06 | 0.13 | 0.17 | 0.50 |
| **ViLBERT** | Standard VQA | 66.98 | 0.19 | 0.17 | 0.21 | 0.89 | 69.23 | 0.20 | 0.18 | 0.21 | 0.99 |
| | VectorScale | 66.87 | 0.14 | 0.13 | 0.19 | 0.65 | 69.04 | 0.17 | 0.15 | 0.20 | 0.77 |
| | **Calibrated (Ours)** | 66.90 | **0.05** | **0.04** | **0.16** | **0.49** | 70.59 | **0.02** | **0.03** | **0.15** | **0.46** |
| | **Uni-VQA (Ours)** | **71.65** | 0.07 | 0.07 | 0.17 | 0.53 | **75.63** | 0.06 | 0.12 | 0.16 | 0.49 |
| **VisualBERT** | Standard VQA | 64.92 | 0.14 | 0.13 | 0.19 | 0.68 | 65.28 | 0.19 | 0.16 | 0.21 | 0.87 |
| | VectorScale | 64.83 | 0.14 | 0.13 | 0.19 | 0.63 | 64.40 | 0.18 | 0.16 | 0.21 | 0.76 |
| | **Calibrated (Ours)** | 65.26 | **0.03** | **0.03** | **0.16** | **0.49** | 67.38 | **0.01** | **0.02** | **0.16** | **0.48** |
| | **Uni-VQA (Ours)** | **70.95** | 0.08 | 0.08 | 0.18 | 0.55 | **74.34** | 0.06 | 0.14 | 0.18 | 0.52 |

calibration techniques, such as the DE framework, to enhance the reliability of VQA models and enable more accurate uncertainty estimates not only for ensuring reliability of the entire `Uni-VQA` framework, but also for effective integration with the LLM model.

### G.3 EFFECTS OF DE-BASED VQA ON REDUCING OVERCONFIDENCE IN INCORRECT PREDICTIONS

Figs. 11 (a) and (b) present histograms of confidence scores for incorrect predictions, respectively, made by the standard, and our DE-based `Calibrated` VQAs. Our proposed method, assigns low confidence scores to the majority of incorrect answers, while the standard VQA produces very high confidence scores for a large number of the incorrect answers. This observation confirms that our DE-based `Calibrated` VQA significantly reduces the overconfidence, by pushing the majority of incorrect answers towards lower confidence scores.

### G.4 COMPREHENSIVE EVALUATION OF CALIBRATION USING ALTERNATIVE CALIBRATION METRICS

ECE (Expected Calibration Error) is a standard metric commonly used to assess model calibration; however, it has several known drawbacks, including sensitivity to binning choices, the inability to capture local miscalibrations effectively, and ignoring the distribution of prediction probabilities within each bin. To comprehensively demonstrate the robustness of our proposed calibration approach in improving the calibration of `TS-VQA` models across various architectures, we additionally evaluate it using alternative calibration metrics: 1) *Adaptive Calibration Error (ACE)*, is an extension of ECE that adaptively determines bin sizes to more accurately capture local miscalibration, 2) *Brier Score* measures the squared difference between predicted probabilities and actual outcomes, assessing both calibration quality and sharpness of probabilistic predictions, and 3) *Negative Log Likelihood (NLL)* quantifies the negative log probability assigned to true outcomes, heavily penalizing confident yet incorrect predictions. These metrics provide complementary perspectives essential for robustly evaluating calibration quality. Table 7 summarizes the results, clearly indicating that our calibration method consistently enhances performance across all evaluated calibration metrics.

### G.5 PERFORMANCE COMPARISON OF LLM VS. TS-VQAS IN VARIOUS CONFIDENCE RANGES

In this experiment we compare the performance of `TS-VQA` models, LLM without answer candidates, and LLMs augmented with answer candidates (2 candidates are given) across all four architectures, for both standard and our `Calibrated` VQA models. We evaluate the performance of these models in terms of accuracy for samples whose confidences, as determined by the respective `TS-VQA` (`Calibrated` or standard), fall within three different confidence ranges: 1) low $(0-0.1)$, 2) moderate $(0.4-0.5)$, 3) and high $(0.95-1)$. Results are presented in Table 8.

Table 8: Comparison between predictive performances of LLM and our `Calibrated TS-VQA` in low- and high-confidence regimes. In low confidence levels, total delegation to LLM yields higher accuracy, while it is misled when presented with the answer candidates from the VQA model. On the contrary, in high confidence levels, VQA model outperforms LLM, suggesting that high-confident questions can be answered in a more efficient manner by the VQA.

| Model | | $c \in [0, 0.1]$ | | | $c \in [0.4, 0.5]$ | | | $c \in [0.95, 1]$ | |
|---|---|---|---|---|---|---|---|---|---|
| | TS-VQA | LLM w. candidates | LLM | TS-VQA | LLM w. candidates | LLM | TS-VQA | LLM w. candidates | LLM |
| Calibrated **Pythia** | 4.6 | 6.97 | **14.23** | 39.41 | 46.29 | **46.83** | **90.95** | 89.79 | 86.85 |
| Calibrated **CLIP-ViL** | 6.95 | 8.58 | **16.17** | 37.01 | **39.45** | 36.52 | **89.88** | 87.94 | 83.64 |
| Calibrated **ViLBERT** | 6.5 | 10.47 | **18.88** | 45.15 | **49.30** | 46.75 | **91.41** | 90.14 | 87.14 |
| Calibrated **VisualBERT** | 7.12 | 13.16 | **20.18** | 47.56 | **54.44** | 52.48 | **93.19** | 92.02 | 90.05 |
| Standard **Pythia** | 3.84 | 6.42 | **12.29** | 32.13 | 35.88 | **37.31** | 83.94 | **84.43** | 81.87 |
| Standard **CLIP-ViL** | 4.37 | 6.34 | **14.10** | 27.42 | 30.66 | **31.91** | 81.50 | 81.03 | 77.14 |
| Standard **ViLBERT** | 2.54 | 5.57 | **12.56** | 26.49 | 30.68 | **33.25** | 79.68 | **81.14** | 77.87 |
| Standard **VisualBERT** | 3.33 | 8.38 | **14.16** | 31.36 | 36.44 | **38.52** | 83.08 | **84.39** | 82.02 |

For our `Calibrated` models, we observe that in the low confidence range, the LLM alone is the most effective. In the moderate confidence range, providing answer candidates from the `TS-VQA` generally improves the performance of the LLM. However, in the high confidence range, the `TS-VQA` outperforms the LLM. This suggests that answering hgih-confidence questions using the `TS-VQA` model, rather than the LLM, not only reduces the burden on the LLM and improves efficiency, but also benefits the hybrid approach in terms of improving the accuracy.

In contrast, when using a standard VQA as the `TS-VQA`, we observe that the LLM achieves the highest accuracy in both the low and moderate confidence ranges. The lower accuracy of the LLM with answer candidates indicates that the provided top-$k$ answer candidates reduces the accuracy as compared to when no candidates are provided, suggesting poorer quality of the answer candidates set.

In the highest confidence range, the LLM with answer candidates generally performs better than both the LLM alone and the `TS-VQA`. This behavior makes the effectiveness of a hybrid approach suboptimal for any delegation confidence threshold when using a standard VQA model.

These findings highlight the importance of calibrating the `TS-VQA` model using the diverse ensemble, as it enables a more effective hybrid approach that leverages the strengths of both the `TS-VQA` and the LLM in different confidence ranges. By delegating low-confidence question to the LLM, incorporating answer candidates for moderate-confidence questions, and relying on the `TS-VQA` for high-confidence questions, our proposed approach improves both accuracy and efficiency in the VQA task.

### G.6 Effectiveness of the Dynamic Top-K Selection

To evaluate the effectiveness of the proposed uncertainty-guided dynamic answer candidate selection, we compare the performance of the Uni-VQA framework against the same approach with fixed top-$K$ answer candidates provided for all confidence levels. In all methods, the `TS-VQA` model is our `Calibrated` VQA, trained according to the diverse ensemble. We refer to these variants as LLM-`Calibrated` (Top-$K$), where $K$ represents the number of answer candidates provided to the LLM model.

Fig. 13 presents the VQA accuracy with respect to the delegation thresholds for various $K$ values, across all 4 architectures. The figures suggest that the dynamic approach, *i.e.,* `Uni-VQA`, achieves the highest overall accuracy for any delegation threshold. Additionally, for any given accuracy, the dynamic approach achieves the lowest delegation percentage among the other variants, while also achieving a higher accuracy than the highest achieved by the fixed top-$K$ answer candidate variants, at certain delegation thresholds.

A comparison between the accuracy of the methods at fixed thresholds for thresholds below $0.2$ highlights the effectiveness of the LLM-only prompting when no answer candidates are provided (Top-0). The VQA accuracies of the LLM-`Calibrated` (Top-1), and LLM-`Calibrated` (Top-10) variants at lower thresholds suggest that providing answer candidates in this region confuses the LLM, compared to when those answer candidates are not present. This can be attributed to the

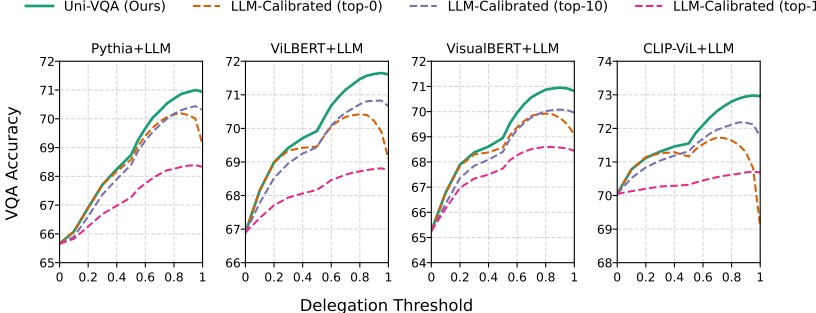

Figure 13: Performance comparison of the proposed `Uni-VQA`, against `LLM-Calibrated` with fixed top-$K$ answer candidates, with respect to the delegation threshold.

answer candidates being random guesses in the low-confidence region, indicating the model's total lack of knowledge.

These findings demonstrate the superiority of the dynamic top-$K$ selection approach employed by `Uni-VQA`. By adaptively selecting the number of answer candidates based on the confidence of the `TS-VQA` model, `Uni-VQA` achieves higher accuracies and lower delegation percentages compared to fixed top-$K$ variants. Furthermore, the results emphasize the importance of relying solely on the LLM for low-confidence questions, as providing answer candidates in this region can hinder the LLM's performance. The dynamic approach effectively leverages the strengths of both the `TS-VQA` and the LLM, leading to improved overall performance in the VQA task.

### G.7 UNI-VQA HYPERPARAMETER GENERALIZABILITY

To validate the robustness of our `Uni-VQA` framework and demonstrate that it does not require careful per-model hyperparameter tuning, we conducted an extensive cross-model hyperparameter transfer analysis. This analysis evaluates whether hyperparameters optimized for one `TS-VQA` backbone can effectively transfer to other architectures without significant performance degradation.

For each VQA model in our framework, we apply the hyperparameters $\{l, u, K(c_i)\}$ originally tuned for that specific model to all other models in our evaluation set. This cross-application tests whether our delegation mechanism maintains consistent performance across architectural variations. The hyperparameters include: (1) the delegation threshold $u$ that determines when to invoke the LLM, (2) The dynamic top-$K$ bounds $(l, u)$ that control answer candidate selection, (3) The confidence-adaptive function $K(c_i)$ that adjusts selection based on model confidence.

**Key Findings.** The analysis reveals remarkable robustness in our hyperparameter selection. The maximum deviation from optimal performance across all cross-model transfers is only 1.24% (CLIP-ViL using BEiT-3 hyperparameters), with most deviations below 0.6%. This demonstrates that hyperparameters are not overly specialized to individual architectures. Additionally, the symmetry in the transfer matrix (e.g., ViLBERT $\rightarrow$ Pythia and Pythia $\rightarrow$ ViLBERT both maintain high accuracy) confirms that the hyperparameter robustness is bidirectional, not dependent on specific source-target model pairs.

These results validate our claim in Section 5.3 of main paper, that the `Uni-VQA` framework is not sensitive to careful hyperparameter tuning, making it a practical and scalable solution for real-world VQA applications. The framework's ability to maintain consistent performance across diverse architectures with shared hyperparameters addresses a critical deployment challenge in VQA systems.

### G.8 ALTERNATIVE UNCERTAINTY MEASURES FOR UNI-VQA: ENTROPY

While our main approach uses confidence scores (*i.e.,* maximum output probability) to guide knowledge exchange between `TS-VQA` and LLM, we also explored an alternative uncertainty measure to assess robustness of our delegation strategy. A natural alternative to confidence score is *entropy*, which is widely used in uncertainty quantification literature (Lakshminarayanan et al., 2017; Kendall

Table 9: Cross-model hyperparameter generalizability on COCO-QA. Each cell shows accuracy when model in the corresponding row uses hyperparameters (HP) tuned for model in the column. Bold values indicate model-specific tuned parameters. Max Dev shows maximum deviation from optimal performance.

| Model Evaluated | HP from CLIP-ViL | HP from Pythia | HP from ViLBERT | HP from VisualBERT | HP from BEIT3 | Max Dev |
|---|---|---|---|---|---|---|
| **CLIP-ViL** | **74.95** | 75.08 | 75.11 | 74.98 | 73.71 | 1.24 |
| **Pythia** | 74.76 | **74.78** | 74.78 | 74.82 | 74.41 | 0.37 |
| **ViLBERT** | 75.57 | 75.61 | **75.63** | 75.55 | 75.06 | 0.57 |
| **VisualBERT** | 74.28 | 74.25 | 74.25 | **74.32** | 73.72 | 0.60 |
| **BEIT3** | 76.08 | 75.99 | 75.99 | 75.90 | **76.01** | 0.11 |

& Gal, 2017), as it provides a measure of the prediction uncertainty by quantifying the dispersion of the probability mass across possible answers.

To empirically evaluate the effectiveness of our approach using "entropy" as an uncertainty measure for delegation and knowledge exchange, we implement an entropy-based delegation variant of `Uni-VQA` and compared it with our confidence-based approach. Table 10 compares performances of our `Uni-VQA` using the two uncertainty-measure, in terms of accuracy and LLM delegation percentage for ViLBERT `TS-VQA` on VQA-v2 dataset, and shows that both uncertainty measures achieve comparable performance, with confidence-based delegation showing slight advantage in both accuracy and efficiency.

Table 10: Performance comparison between confidence-based and entropy-based uncertainty measures for knowledge exchange in `Uni-VQA` using ViLBERT on VQA-v2 dataset.

| **Uncertainty Measure** | ACC(↑) | LLM-Delegation(%)(↓) |
|---|---|---|
| Confidence-based | 71.6 | 79.1 |
| Entropy-based | 71.5 | 80.0 |

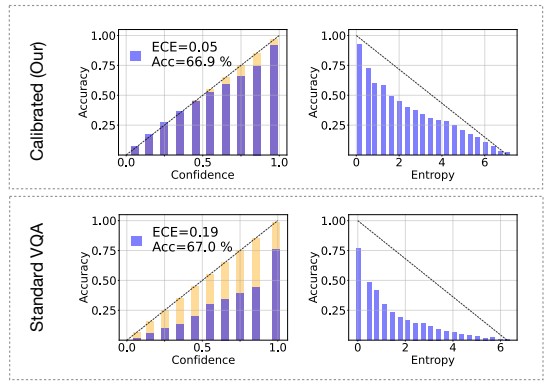

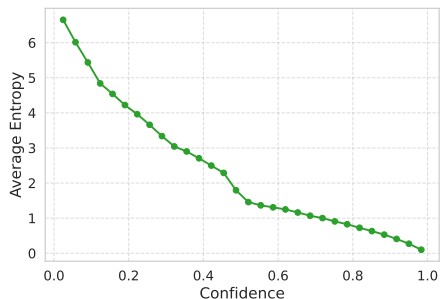

(a) Relationship between accuracy and uncertainty measures (confidence vs. entropy) for `Calibrated` (top) and Standard VQA (bottom) `TS-VQA` models.

(b) Inverse relationship between model prediction confidence and output entropy, averaged over 30 confidence bins.

Figure 14: Visualizations of confidence and entropy relationships in ViLBERT on VQA-v2 dataset.

Fig. 14a illustrates the relationship between both uncertainty measures (entropy and confidence score) and accuracy, for standard and our DE-based `Calibrated TS-VQAs`. For low-entropy (corresponding to high-confidence) regions, our calibrated models consistently achieves higher accuracy compared to high-entropy regions, indicating that the `Calibrated` model's answers are more reliable in low-entropy regions, confirming that both measures effectively identify samples where the $TS-VQA$ model can be trusted. Additionally, Fig. 14b depicts the relationship between average answer confidences and probability entropies, calculated in 30 equally spaced confidence intervals, illustrating a clear inverse trend where higher confidence values correspond to lower entropy in the predicted distributions.

While entropy can also serve as a proxy for uncertainty, we choose confidence as our primary uncertainty measure for several practical reasons: (1) **Interpretability**: Confidence is bounded between 0 and 1 with an intuitive probabilistic interpretation, where in a well-calibrated model confidence of 0.9 suggests a 90% probability of correctness. On the contrary, entropy ranges between 0 and $\log_2(C)$ where $C$ is the number of answer classes, which makes setting and interpreting thresholds less intuitive. (2) **Direct relationship with calibration:** confidence score is a widely used measure in calibration literature. Additionally, calibration metrics including $ECE$ are specifically designed to measure the alignment between confience scores and accuracies (both bounded between 0 and 1), hence confidence score is a natural choice for our framework. (3) **Simplicity:** Using confidence scores for both calibration assessment and delegation decisions leads to a simpler framework. Also, confidences are straightforward to obtain, while computing entropy introduces additional computational overhead.

### G.9 DIVERSE ENSEMBLE DISTILLATION

The `Uni-VQA` framework is designed to reduce overall computational costs by reducing dependence on large-scale LLM models. Although effective during the inference phase, the use of an ensemble model increases the computational costs of inference by `TS-VQA` and may potentially lead to higher latency. To address this issue, inspired by the findings of Allen-Zhu & Li (2023); Hebbalaguppe et al. (2024) on the advantages of ensemble learning and knowledge distillation to transfer predictive accuracy and calibration, we use knowledge distillation to transform the calibrated diverse ensemble model (DE) into a single VQA model, with the same architecture as individual ensemble components, and is trained to learn from the ensemble's output distribution instead of the target labels, thereby preserving both the ensemble's accuracy and enhanced calibration.

The distillation process minimizes the Kullback-Leibler divergence between the output distributions of the ensemble and the distilled model, expressed as follows:

$$\mathcal{L}_{KD}(X; \theta_s) = T^2 \sum_{i=1}^{N} \text{KL}\left(\sigma\left(\frac{f_s(x_i)}{T}\right) || \sigma\left(\frac{f_t(x_i)}{T}\right)\right),$$

where $T$ is the temperature parameter used to smooth the probability distributions, and $\sigma$ represents the softmax function. This process enables the distilled model to retain the ensemble's strengths while reducing the operational costs associated with deploying multiple models.

**Accuracy & Calibration Performance Preservation.** Table 11 highlights the effectiveness of this approach in preserving calibration and predictive performance across four VQA architectures, on VQA-v2 and COCO-QA datasets. The distilled model maintains the accuracy and calibration properties of the DE, while significantly reducing the computational overhead associated with ensembled models. As shown in Table 12, the increased inference time caused by ensembling is effectively remedied when the ensemble model is distilled into a single VQA model. These latency measurements were obtained by running models on a single A6000 GPU, with a batch size of 32, averaged over 3 runs.

**Integration with Uni-VQA Framework.** While all results presented in the main paper utilize the original ensemble models, we validate that distilled models can serve as efficient alternatives within the `Uni-VQA` framework. Table 13 presents a direct comparison on the COCO-QA dataset, showing that distilled models not only maintain comparable accuracy but also achieve more efficient delegation patterns. Specifically: (1) VilBERT and VisualBERT demonstrate $5 - 6\%$ reduction in LLM delegation while slightly improving accuracy (+0.26% and +0.39% respectively), indicating enhanced confidence in local question answering. (2) CLIP-ViL maintains robust performance with minimal change in delegation behavior (+0.55%), preserving the ensemble's already efficient delegation pattern.

### G.10 EVALUATION ON RECENT TRANSFORMER-BASED VQA ARCHITECTURES

We extend our evaluation to include ViLT (Vision-and-Language Transformer) (Kim et al., 2021), a state-of-the-art transformer-based model that represents recent advances in vision-language understanding. Unlike the earlier architectures evaluated in our main experiments (VisualBERT, ViLBERT, CLIP-ViL), ViLT employs a simpler design that processes raw image patches directly through

Table 11: Performance comparison of diverse ensemble and distilled VQA across four architectures.
*Diverse Ensemble requires *three times* the total parameters of Distilled VQA since it comprises *three* independently trained models.

| | Model | Diverse Ensemble* | | Distilled VQA | |
|---|---|---|---|---|---|
| | | ACC↑ | ECE↓ | ACC↑ | ECE↓ |
| **VQA-v2** | **Pythia** | 66.15 | 0.06 | 65.92 | 0.05 |
| | **CLIP-ViL** | 70.05 | 0.07 | 69.64 | 0.07 |
| | **ViLBERT** | 66.90 | 0.05 | 67.29 | 0.05 |
| | **VisualBERT** | 65.26 | 0.03 | 65.40 | 0.03 |
| **COCO-QA** | **Pythia** | 68.64 | 0.02 | 68.66 | 0.02 |
| | **CLIP-ViL** | 69.94 | 0.02 | 70.03 | 0.03 |
| | **ViLBERT** | 70.59 | 0.02 | 70.43 | 0.03 |
| | **VisualBERT** | 67.38 | 0.01 | 67.94 | 0.02 |

Table 12: Average inference latency (ms) comparison between the Diverse Ensemble (DE), and the distilled VQA model.

| Model | Average Latency (ms) | |
|---|---|---|
| | Diverse Ensemble | Distilled VQA |
| **Pythia** | 4.29 | 3.71 |
| **CLIP-ViL** | 59.94 | 24.0 |
| **ViLBERT** | 18.51 | 9.84 |
| **VisualBERT** | 15.49 | 9.61 |

Table 13: Performance and delegation comparison of diverse ensemble and distilled `TS-VQA` models on COCO-QA dataset, showing accuracy and LLM delegation percentages.
*Diverse Ensemble column corresponds to the results in the main paper presented in Table 1., where Distilled VQA corresponds using the Distilled model as the `Calibrated TS-VQA`.

| Model | Diverse Ensemble | | Distilled VQA | |
|---|---|---|---|---|
| | ACC↑ | Deleg %↓ | ACC↑ | Deleg %↓ |
| **ViLBERT** | 75.63 | 67.19 | **75.89** | **61.68** |
| **VisualBERT** | 74.34 | 73.46 | **74.73** | **67.66** |
| **CLIP-ViL** | 74.95 | **64.89** | **75.05** | 65.44 |

a transformer, without relying on pre-extracted region features, making it more representative of modern end-to-end vision-language architectures.

We train our Calibrated ViLT models using the same diverse ensemble configuration as other architectures, with DRO hyperparameters $\lambda \in \{2, 3, 4\}$ for COCO-QA and $\lambda \in \{8, 20, 100\}$ on VQA-v2. All other training hyperparameters follow the original ViLT implementation.

Table 14 presents a comprehensive comparison of Standard ViLT, our Calibrated ViLT, and Uni-VQA integration on COCO-QA, alongside the LLM-only baseline. The results demonstrate that our diverse ensemble approach effectively improves calibration for modern transformer architectures. Our Calibrated ViLT achieves substantial calibration improvements, reducing ECE from **0.17** to **0.02**, while maintaining comparable accuracy to Standard ViLT. Furthermore, when integrated into the Uni-VQA framework, ViLT achieves the highest accuracy of (76.33% on COCO-QA and 74.31% on VQA-v2) and efficient LLM delegation of (70.47% on COCO-QA and 65.31% on VQA-v2), demonstrating that our approach maintains its effectiveness on modern transformer-based architectures.

### G.11    ROBUSTNESS TO DISTRIBUTION SHIFT AND OUT-OF-DISTRIBUTION GENERALIZATION

A critical concern for real-world VQA deployment is whether calibrated confidence scores remain reliable under distribution shifts, or out-of-distribution questions. To evaluate the robustness of our calibration approach, we conduct experiments on the AdVQA dataset (Sheng et al., 2021), an adversarial out-of-distribution benchmark specifically designed to challenge VQA model robustness through carefully constructed adversarial question-answer pairs.

**Experimental Setup:** We evaluate VQA models trained on VQA-v2 directly on the AdVQA test set without any finetuning, creating a true out-of-distribution evaluation scenario. This setup tests whether our diverse ensemble calibration maintains its advantages when facing distribution shifts that differ from the training distributions. We compare Standard VQA models (trained with cross-entropy loss) against our `Calibrated` models across four architectures: Pythia, CLIP-ViL, ViLBERT, and VisualBERT.

**Out-of-Distribution Calibration Performance:** Table 15 presents the performance of Standard and `Calibrated` VQA models on the AdVQA dataset. As expected, all models experience significant accuracy degradation and increased calibration error compared to in-distribution performance (Table 1). However, the critical finding is that our `Calibrated` models consistently maintain better calibration than Standard models across all architectures.

Table 14: Performance comparison of `Uni-VQA` with `TS-VQA` models and LLM on ViLT architecture.

| | Model | VQA-v2 | | | COCO-QA | | |
|---|---|---|---|---|---|---|---|
| | | ACC↑ | ECE↓ | LLM-Deleg(%)↓ | ACC↑ | ECE↓ | LLM-Deleg(%)↓ |
| | LLM-only (Mistral-7B) | 69.09 | 0.31 | 100 | 72.03 | 0.27 | 100 |
| **ViLT** | Standard VQA | 66.60 | 0.21 | - | 73.61 | 0.17 | - |
| | **Calibrated** (Ours) | 66.44 | **0.07** | - | **73.89** | **0.02** | - |
| | **Uni-VQA** (Ours) | **74.31** | **0.04** | 65.31 | **76.33** | 0.03 | 70.47 |

Table 15: Out-of-distribution performance comparison on AdVQA dataset (test split). All models are trained on VQA-v2, and evaluated on AdVQA to assess robustness of our `Calibrated` and `Uni-VQA` models under distribution shift.

| | Model | VQA-v2 | | | AdVQA | | |
|---|---|---|---|---|---|---|---|
| | | ACC↑ | ECE↓ | LLM-Deleg(%)↓ | ACC↑ | ECE↓ | LLM-Deleg(%)↓ |
| | LLM-only (Mistral-7B) | 69.09 | 0.31 | 100 | 38.98 | 0.53 | 100 |
| **Pythia** | Standard VQA | 65.67 | 0.14 | - | 30.6 | 0.36 | - |
| | **Calibrated** (Ours) | 66.15 | 0.06 | - | 31.5 | **0.12** | - |
| | **Uni-VQA** (Ours) | **71.00** | **0.05** | 78.77 | **41.05** | 0.19 | 98.08 |
| **CLIP-ViL** | Standard VQA | 69.95 | 0.18 | - | 32.13 | 0.23 | - |
| | **Calibrated** (Ours) | **70.05** | 0.08 | - | 31.95 | **0.06** | - |
| | **Uni-VQA** (Ours) | **72.98** | **0.07** | 69.86 | **38.11** | 0.11 | 99.86 |
| **ViLBERT** | Standard VQA | 66.98 | 0.19 | - | 32.36 | 0.37 | - |
| | **Calibrated** (Ours) | 66.90 | **0.05** | - | 32.07 | **0.20** | - |
| | **Uni-VQA** (Ours) | **71.65** | 0.07 | 79.06 | **40.21** | 0.19 | 91.76 |
| **VisualBERT** | Standard VQA | 64.92 | 0.14 | - | 31.41 | 0.36 | - |
| | **Calibrated** (Ours) | 65.26 | **0.03** | - | 31.53 | **0.14** | - |
| | **Uni-VQA** (Ours) | **70.95** | 0.08 | 77.87 | **40.77** | 0.16 | 96.03 |

Notably, we observe that across all architectures, our `Calibrated` models achieve lower ECE compared to Standard models on AdVQA. This demonstrates that the calibration benefits of diverse ensemble training are not limited to in-distribution data. While all methods exhibit higher ECE on AdVQA compared to VQA-v2, which is expected behavior under distribution shift, the relative advantage of our calibration approach remains consistent.

**Analysis of Confidence Distribution Under Distribution Shift:** To get insights on how calibration behaves at a more granular level under distribution shift, we analyze the confidence distributions of correct and incorrect predictions on AdVQA. Figs. 15 and 16 present confidence histograms comparing Standard and our `Calibrated` ViLBERT models.

As illustrated in Fig. 15, Standard VQA exhibits severe overconfidence on incorrect predictions, with a pronounced spike in the highest confidence bin (around 1.0), indicating that the model is overconfident on many incorrect answers. In contrast, our `Calibrated` model shifts the distribution of incorrect predictions toward lower confidence regions, with substantially higher concentration in the low-confidence bins (particularly in the 0.0-0.3 range). The overconfident spike at confidence 1.0 is greatly reduced in the `Calibrated` model. These patterns mirror the in-distribution behaviors observed in Figs. 11, 12, confirming that diverse ensemble training continues to shift incorrect predictions to lower confidence regions even under distribution shift.

**Implications for RAG integration:** The above analysis reveals an important opportunity for integrating Retrieval-Augmented Generation (RAG) methods (Lewis et al., 2020; Guu et al., 2020) with `Uni-VQA`. Our diverse ensemble calibration reliably pushes OOD and knowledge-intensive questions toward the lowest-confidence region (precisely the regime where the LLM is invoked without `TS-VQA` candidates). To further validate this, we evaluated our calibrated `TS-VQA` (trained on VQA-v2) on OK-VQA (Marino et al., 2019), a knowledge-based VQA benchmark requiring external world knowledge. On OK-VQA, $56.6\%$ of incorrect predictions from our calibrated model (accuracy: $20\%$, ECE: 0.11) fall below the lower confidence threshold, compared to only $27\%$ for standard `TS-VQA`. This confirms that knowledge-heavy questions are reliably routed to the lowest-confidence region. In `Uni-VQA`, this is exactly where RAG augmentation could be most beneficial—enhancing LLM accuracy on knowledge-intensive questions while avoiding the cost of invoking RAG on every query.

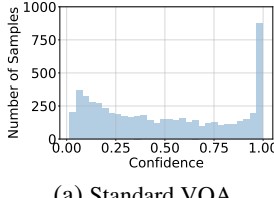 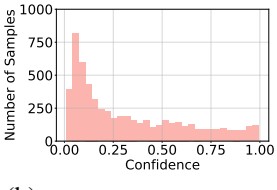

(a) Standard VQA         (b) `Calibrated(Ours)`

Figure 15: Confidence histograms for *incorrect* predictions on AdVQA (OOD; trained on VQA-v2): (a) Standard VQA and (b) `Calibrated` (ours).

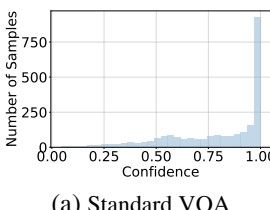 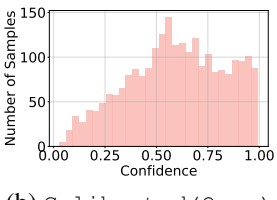

(a) Standard VQA         (b) `Calibrated(Ours)`

Figure 16: Confidence histograms for *correct* predictions on AdVQA (OOD; trained on VQA-v2): (a) Standard VQA and (b) `Calibrated` (ours).

## G.12 COMPREHENSIVE COMPUTE COST ANALYSIS

To provide a complete picture of our framework's efficiency, we present a detailed breakdown of both training and inference costs. While training introduces upfront computational overhead, the significant inference savings in production deployments justify this initial investment.

Our analysis considers three primary computational costs: (1) **Training Cost:** One-time GPU hours required for ensemble model training, (2) **Distillation Cost:** Additional training to compress ensembles (optional), (3) **Inference Cost:** Per-sample latency at inference time.

Table 16 presents the costs associated with training of our `Uni-VQA` components. If distillation is employed (optionally), it adds approximately one-third of the ensemble training time (equivalent to training a single model). Table 17 presents the effective inference costs in our hybrid system, accounting for selective delegation.

**Key Cost-Benefit Findings.** Inference costs dominate real-world computational expenses in production systems. While training the ensemble models requires an upfront investment of 15-366 GPU hours depending on the chosen backbone, this cost is quickly amortized in production deployments that process millions of queries daily. For instance, at a scale of 10 million queries per day, our framework's improved inference efficiency translates to savings of approximately 11,000-35,000 GPU hours monthly compared to LLM-only inference.

Table 16: Comprehensive compute cost breakdown for `Uni-VQA` components on VQA-v2 dataset. Calibrated models use ensemble of 3 independently trained models. Training time measured on A100 GPUs, inference latency on A6000 GPU.

| Model | Parameters (M) | Training Time (GPU Hours) | Avg Inference Time (ms/sample) |
|---|---|---|---|
| *Calibrated Models (Ensemble of 3)* | | | |
| Pythia | $3 \times 147$ | $3 \times 5 = 15$ | $3 \times 3 = 9$ |
| ViLBERT | $3 \times 250$ | $3 \times 47 = 141$ | $3 \times 9 = 27$ |
| VisualBERT | $3 \times 114$ | $3 \times 19 = 57$ | $3 \times 9 = 27$ |
| CLIP-ViL | $3 \times 256$ | $3 \times 122 = 366$ | $3 \times 23 = 69$ |
| BEiT-3 | $3 \times 1,900$ | $3 \times 72 = 216$ | $3 \times 9 = 27$ |
| *Reference: LLM-only Baseline* | | | |
| Mistral-7B | 7,000 | 0* | 534 |

*Pre-trained model used without additional training

Table 17: Effective inference latency comparison between `Uni-VQA` and LLM-only baseline. Delegation % indicates frequency of LLM invocation. Effective latency computed as: $t_{VQA} + (\frac{\text{Deleg}\%}{100} \times t_{LLM})$.

| TS-VQA Backbone | TS-VQA Latency (ms) | Delegation % | Effective Latency (ms) | Speedup vs LLM-only |
|---|---|---|---|---|
| Mistral-7B only | — | 100% | 534 | 1.00× |
| Pythia | 9 | 78.8% | 426 | 1.25× |
| ViLBERT | 27 | 79.1% | 447 | 1.19× |
| VisualBERT | 27 | 77.9% | 440 | 1.21× |
| CLIP-ViL | 69 | 69.9% | 440 | 1.21× |
| BEiT-3 | 27 | 35.9% | 217 | 2.46× |

Table 18: Hyperparameters for training our `Calibrated` VQA models.

| | VQA Model | $\lambda_1$ | $\lambda_2$ | $\lambda_3$ |
|---|---|---|---|---|
| **VQA-v2** | Pythia | 8 | 100 | 1000 |
| | ViLBERT | 8 | 20 | 100 |
| | VisualBERT | 10 | 20 | 100 |
| | CLIP-ViL | 20 | 100 | 1000 |
| | BEiT-3 | 8 | 200 | 500 |
| **COCO-QA** | Pythia | 2 | 4 | 200 |
| | ViLBERT | 2 | 3 | 4 |
| | VisualBERT | 2 | 3 | 5 |
| | CLIP-ViL | 1 | 2 | 50 |
| | BEiT-3 | 0.05 | 0.5 | 5 |

### G.13 EXPERIMENTS REPRODUCIBILITY

In this section the hyperparameters used for training the Diverse ensemble based `Calibrated` model. The DRO loss can be computationally expensive to optimize. To mitigate this, similar to the approach in (Sapkota et al., 2024), we employ a regularized version of the loss function, defined as:

$$\mathcal{L}(\Theta)^{DRO} = \max_{\mathbf{w}, \mathbf{w}^T \mathbb{1}=1} \sum_{n=1}^{N} w_n l(\mathbf{x}_n, \Theta) - \lambda D_f \left( \mathbf{p} \parallel \frac{\mathbb{I}}{N} \right), \quad (21)$$

which has a closed-form solution as demonstrated in (Sapkota et al., 2024):

$$\mathcal{L}(\Theta)^{DRO} = \sum_{n=1}^{N} w_n^* l(\mathbf{x}_n, \Theta) \quad (22)$$

where, $w_n^*$ is given as

$$w_n^* = \frac{\exp(\frac{l(\mathbf{x}_n, \Theta)}{\lambda})}{\sum_{j=1}^{N} \exp(\frac{l(\mathbf{x}_j, \Theta)}{\lambda})} \quad (23)$$

In this setup, our hyperparameters are the $\lambda$ values corresponding to the diverse models in the ensemble. For all of our experiments, we set the ensemble count to 3, resulting in three hyperparameters: $\lambda_1$, $\lambda_2$, and $\lambda_3$. For training our `Calibrated TS-VQA` models. We use $\lambda \in \{8, 10, 20, 50, 100, 200, 500, 1000\}$ in our experimentation, and select the final parameters based on the performance on the validation set, to obtain the desired ECE. The final values of hyperparameters are given in Table 18. Due to computational overhead of LLM-based inferences, we report results based on a single run.

## H QUALITATIVE ANALYSIS

Fig. 17 presents qualitative examples illustrating how `Uni-VQA` performs confidence-guided knowledge exchange between `TS-VQA` and the LLM. For each example, we show the input image–question pair, the `TS-VQA` model's initial prediction with its confidence score (spanning low to high confidence bins), and the top answer candidates proposed by `TS-VQA`. We then report the LLM's prediction (and its correctness) when prompted with different numbers of answer candidates, reflecting the behaviors discussed in section 3.3.

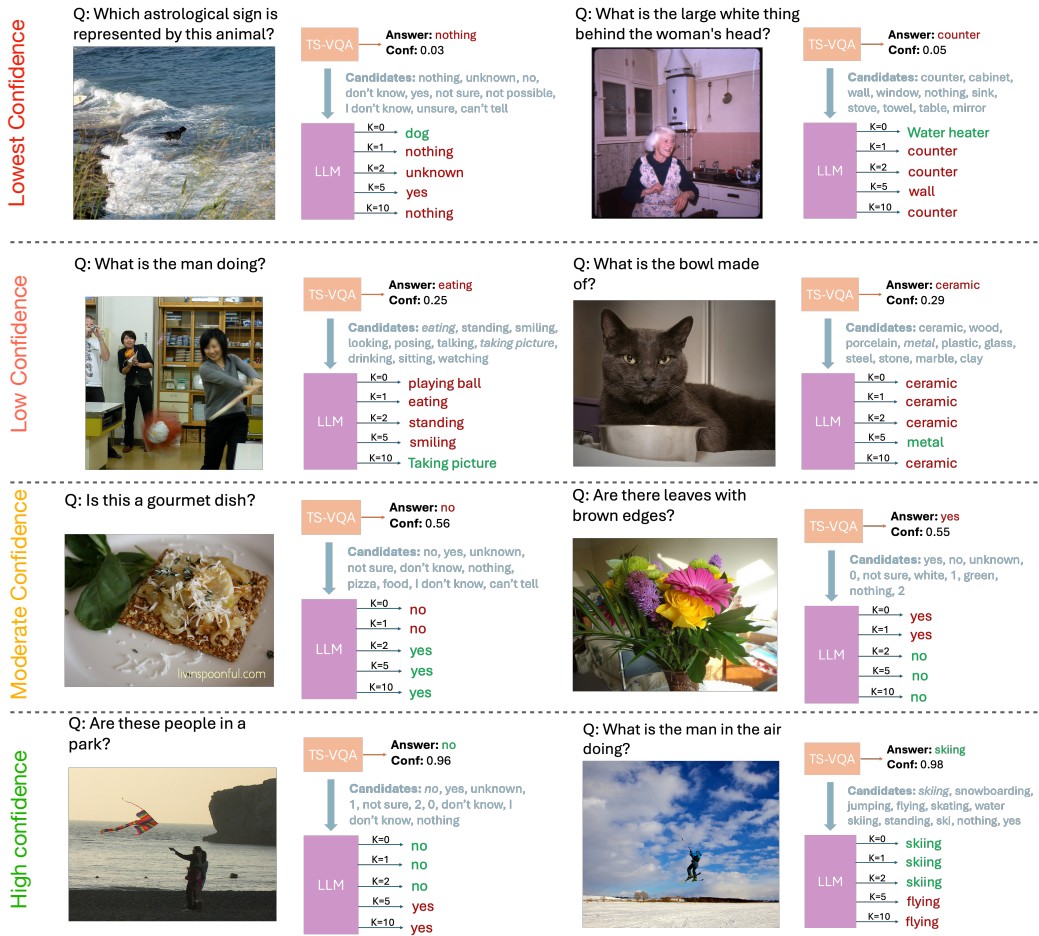

Figure 17: Qualitative examples demonstrating the knowledge exchange in various confidences of TS-VQA

These examples reveal a consistent trend across confidence regimes. In the lowest-confidence bin, both the TS-VQA prediction and its candidate set are often misleading; consequently, providing these candidates can steer the LLM toward an incorrect answer. In this regime, prompting the LLM without answer candidates (top-0) frequently yields the correct response. In the low-confidence regime, the LLM benefits from receiving a larger candidate set (e.g., top-10), which can provide useful partial signal despite the TS-VQA's uncertainty. As confidence increases, fewer candidates become sufficient (and can be preferable) since the top candidates are more reliable and additional candidates may introduce noise. Finally, in the highest-confidence regime, where the calibrated TS-VQA prediction is most reliable, the system can accept the TS-VQA answer directly without delegating to the LLM, saving expensive LLM computation while maintaining accuracy.

## I    BROADER IMPACT STATEMENT

Modern large language model (LLM)-based systems have revolutionized AI applications, demonstrating remarkable capabilities in diverse domains, including healthcare, finance, and creative industries. Yet their widespread adoption comes at a substantial environmental cost, raising concerns about sustainability and their environmental impacts. Studies (Strubell et al., 2020; Patterson et al., 2021) have highlighted the environmental costs of training and deploying these models, highlighting the significant carbon footprint associated with large-scale AI, emphasizing on the need for more energy efficient AI solutions. Furthermore, reports (Patterson et al., 2021; Weidinger et al., 2022;

Luccioni et al., 2024) indicate that inference accounts for a substantial AI workloads, often exceeding the energy costs of model training and development, due to their usage at scale. This underscores the urgent need to develop AI systems that balance computational efficiency with performance.

In line with the principles of Green AI (Schwartz et al., 2020) - prioritizing innovation while minimizing resource consumption and computational costs - our work proposes a framework that selectively and dynamically utilizes LLMs when their unique capabilities are truly needed. Our approach identifies opportunities to use smaller, task-specific models for routine tasks while reserving resource-intensive LLMs for complex queries that demand their advanced capabilities. This selective deployment strategy can significantly reduce the environmental footprint of AI systems without compromising their performance.

While our approach improves trustworthiness through calibration, and efficiency of using LLMs by reducing overreliance on the LLMs, several negative aspects merit further discussion. Firstly, calibrated confidence scores are critical in domains like medical, autonomous driving, or surveillance, where incorrect answers can have serious consequences. Although our framework improves reliability, ***a high model confidence does not guarantee correctness***, and in such high-stakes scenarios, a human supervision must make an informed decision. If such confidence scores are interpreted as definitive indicators of correctness (especially by non-expert users), this could lead to overtrust and potential harmful decisions in sensitive contexts. Secondly, our framework involves dynamic delegation of queries to LLMs, which may reside in third-party systems. In scenarios involving sensitive or private visual data, delegation to an external LLM (particularly one not hosted locally), poses serious privacy risks. Moreover, unless made explicitly transparent to users when delegation occurs, this can lead to unintended data exposure and ethical concerns around informed consent.

## J  LIMITATIONS AND FUTURE WORK

Our study has several limitations. First, while our approach employs confidence-based delegation from TS-VQA to the LLM with answer candidates, it does not leverage additional mechanisms, such as answer consistency checking or refinement techniques (Srinivasan et al., 2024; Khan et al., 2024; Prasad et al., 2023), which could further boost the performance, when answering is delegated to an LLM. Second, our approach still lacks the systematic way of providing the well-calibrated uncertainty estimates on the LLM-generated answers. While calibrated confidence estimates of our `Calibrated TS-VQA` provides a better reflection on the question difficulty, accurate confidence estimation of the LLM-generated answers can be important, particularly in safety-critical domains such as medical, or security surveillance. As uncertainty quantification in LLMs remains an ongoing research challenge, we leave the development of more robust LLM calibration strategies for future work.

## K  SOURCE CODE

The source code is available at: `https://github.com/mahmozaffari/Uni-VQA`.

