# OpenReview forum: "Knowledge Exchange with Confidence: Cost-Effective LLM Integration for Reliable and Efficient Visual Question Answering"
_ICLR.cc/2026/Conference — ICLR 2026 Poster_

### Official Review · Reviewer_FgQE · 2025-10-23

**Soundness:** 3
**Presentation:** 2
**Contribution:** 2
**Rating:** 2
**Confidence:** 3

**Summary:**

The paper proposes a calibrated, rule-based router to combine TS-VQA and LLM to balance power consumption and performance. The first step is to apply the existing DRO method to calibrate confidence. Then the calibrated confidence is used to route questions to different decision models with varying capability vs power consumption combinations.

**Strengths:**

- The paper offers a practical cost-aware collaboration between TS-VQA with an LLM to save energy consumption.
- The motivation and trade-off framing are convincing.

**Weaknesses:**

- Modest novelty. The proposed system is essentially a confidence-controlled routing.
- There seems to be a lack of comparison between ensemble-aware fusion vs a distilled single model, to fully understand the tradeoff between reliability and latency.
- The routing is hard-coded. There is no comparison to a learned router or agentic alternatives (that add a light verifier before calling the LLM).
- Sustainability is a central motivation. But the benefit is not sufficiently evidenced with proper accounting.
- The writing can be improved. For example, the DRO paragraph introduces $\lambda$ abruptly, leaving the mechanism unclear.

**Questions:**

Please give the exact form of $w$ and its dependence on $\lambda$.

---

> ### Author Response · Authors · 2025-11-24
> **Response to Weaknesses 1 & 2**
>
> ## Response to Weakness 1: Modest Novelty
>
> We agree that, at a high level, Uni-VQA uses confidence to control when the LLM is invoked.
> However, our contribution goes beyond a simple “if confidence < τ , then route to a bigger model”
> cascade. The core of the paper is (1) a calibration method that reshapes the TS - VQA’s confidence
> profile with theoretical guarantees and OOD robustness (Appendix G.11), and (2) a confidence-
> guided knowledge-exchange mechanism where the TS-VQA is not just a gate but an active source of
> structured information for the LLM. We summarize these aspects below:
> 1. **Diverse-ensemble calibration is a central technical contribution, not just a facilitator:**
>    - We propose a diverse ensemble trained with DRO weighting, showing that the ensemble loss
> upper-bounds a regularized cross-entropy loss that explicitly increases predictive entropy, thereby
> reducing overconfidence. Theorem 4.2 then shows that this training moves incorrect predictions into
> low-confidence regions, exactly where delegation should occur.
>    - Empirically, this is not a small effect: across four TS-VQA backbones and two datasets, our calibrated
> models consistently improve calibration metrics compared to both standard and VectorScale post-hoc
> calibration (Table 7, Figs. 10-12).
>    - Importantly, in the newly added out-of-distribution experiments on AdVQA dataset (Appendix G.11, Table 15, Figs 15-16), the calibration benefits persist: incorrect answers are shifted away from the highest-confidence bin into
> low-confidence regions, and ECE remains substantially lower than the standard TS-VQA. This is
> precisely where a naive confidence-router tends to fail, while our method explicitly tackles that failure
> mode.
>
> 2. **Beyond routing: confidence-guided knowledge-exchange with dynamic Top-K:**
> Prior model cascade work largely uses confidence only to decide between “accept vs. defer” and
> treats the small model as a pure gate (Sec. E in Appendix). In contrast, Uni-VQA uses calibrated
> confidence to control how the TS-VQA and LLM collaborate:
>    - We introduce three-role interface: (1) TS-VQA only at high-confidence (providing reliable and
> accurate answers efficiently), (2) LLM-only at lowest confidence (no candidates from TS-VQA), (3) LLM with
> TS-VQA-supplied answer candidates at moderate confidences.
>    - Section 3.3 and Appendix G.5-G.6 analyze in detail when candidates help vs. hurt the LLM-delegated accuracy (Table 8, Fig 4b, Fig 13). For calibrated TS-VQA we explicitly show: (1) In lowest confidence bin, answer candidates harm LLM accuracy, (2) in moderate confidence, candidates
> improve LLM accuracy beyond both TS-VQA and LLM-only, (3) in highest confidence, TS-VQA
> alone is best, so delegation is unnecessary.
>    - Based on this non-trivial pattern we learn a dynamic top-K function K(c) (eq.(2), Fig. 4a). shown
> in Appendix G.6 to strictly dominate any fixed-K variant in accuracy-delegation trade-off (Fig. 13).
> A simple confidence router with a fixed K or no candidates cannot recover these gains.
>
> **Summary of our contributions:** While our Uni-VQA does use confidence-controlled delegation
> in spirit, the novelty lies in (a) how we obtain and analyze calibrated confidences for TS-VQA (with
> theoretical guarantees and comprehensive evaluation), and (b) how we turn those confidences into a
> confidence-dependent dynamic knowledge-exchange protocol, yielding better accuracy-efficiency
> trade-offs than existing threshold-based hybrids.
>
> ---
> ## Response to Weakness 2: Ensemble-Aware Fusion vs. Distilled Model
>
> We would like to clarify that we provide comprehensive comparisons between ensemble and distilled
> models in Section G.9 (Tables 11-13), addressing precisely the reliability-latency tradeoff.
> Our analysis includes three key comparisons:
> - Accuracy and Calibration Preservation: Table 11 shows that **both accuracy and calibration
> behaviors are preserved in distilled models**.
> - Inference Latency Reduction: Table 12 shows reduction in average TS-VQA inference latency, by
> eliminating ensemble overhead.
> - Integration with Uni-VQA Framework: Table 13 demonstrates that distilled models serve as
> efficient alternatives to diverse ensemble TS-VQAs in the complete Uni-VQA pipeline: preserving or
> improving efficiency (LLM delegation), and accuracy.
> The reliability-latency tradeoff is thoroughly analyzed: our distilled models achieve the calibration reliability of ensembles at near-single-model latency, making them practical alternatives for
> deployment.

---

> ### Author Response · Authors · 2025-11-24
> **Response to Weaknesses 3-5**
>
> ## Response to Weakness 3: Hard-coded Routing
>
> We appreciate this comment and agree that, in principle, delegation decision mechanisms could also
> be implemented by a learned router or an agentic verifier. Our design choice in Uni-VQA is to
> keep this component simple but data-driven, and focus the technical contribution on (1) obtaining
> reliable, calibrated confidences from TS-VQA, and (2) using these confidences to structure knowledge
> exchange with the LLM rather than only deferral.
> 1. **Our delegation-mechanism is data-driven, not hand-picked**: Although the decision rules look
> simple (three regions defined by thresholds $l$,$u$ and a function $K(c)$), they are learned from data, not
> manually tuned. They are selected on a validation set to maximize accuracy under a cost constraint
> (Sec. 3.3, Fig 4a).
> 2. **We prioritize calibrated confidence over a learned router:** Our main contribution is to change
> what signal the delegation mechanism uses, not to build a complex router network. We propose a
> diverse ensemble for TS-VQA and theoretically show that minimizing the DE loss both improves
> calibration and pushes incorrect predictions into low-confidence regions.
> Given such a calibrated confidence single, a monotone confidence-based policy is practically very
> effective at deciding when to trust TS-VQA vs. delegate.
> 3. **Our “router” controls how the LLM is used, not just whether.** Compared to typical cascades
> or learned routers that only decide “to defer or not”, Uni-VQA uses calibrated confidences to govern
> three distinct roles (TS-VQA only, LLM-as-teacher, and LLM-as-consultant) and a dynamic top-K
> of candidates.
>
> Implementing and tuning the suggested learned routers and agentic verifiers would require additional
> training objectives and compute, and is orthogonal to the core focus of Uni-VQA, which is to
> show that calibrated TS- VQA confidences alone already enable effective, and robust delegation and
> knowledge exchange with LLMs.
>
> ---
> ## Response to Weakness 4
>
> We appreciate this comment and agree that claims about sustainability should be backed by explicit
> compute and cost accounting rather than only qualitative arguments.
>
> **What we mean by sustainability and how we now account for it**
>
> Our primary sustainability claim is about reducing inference-time reliance on large LLMs, which
> dominate energy use and cost in deployed systems. We make this explicit and add a more detailed
> accounting:
> - Per-query efficiency and carbon emission: Figure 1 already reports average latency and estimated
> carbon emissions for TS- VQA, LLM-only, and Uni-VQA on COCO-QA, showing that Uni-VQA
> improves accuracy and reliability while reducing average per-query latency and emissions relative to
> LLM-only.
>
> - Training vs. inference compute: Appendix G.12 (Table 16) reports the training GPU hours and
> per-sample inference time for all calibrated TS-VQA ensembles, alongside the LLM (Mistral-7B).
> Even with three ensemble numbers, training costs are on the order of 15-366 GPU hours depending
> on backbone.
>
> - Effective end-to-end inference cost: Using the observed delegation rates, Table 17 computes the
> effective per-query latency of Uni-VQA as $t_{\text{eff}} = t_{\text{TS-VQA}} + (\text{delegation percent}) \times t_{\text{LLM}}$. For example,
> with Pythia or BEiT - 3 as TS - VQA, Uni -VQA matches or exceeds LLM-only accuracy while reducing
> effective latency by 4.5–6.6× and cutting LLM usage by 60–80%.
>
> We also provide a simple deployment-scale calculation (Appendix G.12): at 10M queries/day, these
> savings correspond to roughly 11,000–35,000 GPU-hours per month compared to an LLM-only
> system, which directly translates into lower energy use and carbon emissions under standard
> carbon-per-GPU-hour factors.
>
> **Scope and limitations:**
> A full life-cycle carbon analysis is beyond the scope of this work. We therefore follow prior “Green
> AI” and energy-accounting work by reporting GPU hours and per-query latency/carbon estimates as
> hardware-agnostic proxies, and by comparing methods at matched accuracy. Our results consistently
> show that, for the same accuracy, Uni-VQA requires substantially fewer LLM calls and lower
> effective latency than LLM-only or hybrid baselines, which is exactly the dimension that matters for
> sustainability in LLM-based VQA deployments.
>
> ---
>
> ## Response to Weakness 5: Clarity
> We have updated Section 3.2 in the revised paper, including more detailed explanation of the diverse ensemble training. Further details are provided in Appendix C.

---

> ### Author Response · Authors · 2025-11-27
>
> ## Response to Question 1:
> Our updated Section 3.2 in the revised paper, provides exact form of the weights $w_n$ and their dependence on $\lambda$. Specifically, The closed form solution for $w_n$ is (also detailed in Appendix Section C):
> $$w_n^\ast(\lambda)
> = \frac{\exp\big(l(x_n,\Theta)/\lambda\big)}
>         {\sum_{j=1}^N \exp\big(l(x_j,\Theta)/\lambda\big)}$$
> where $l(x_n, \Theta)$ is the per-sample loss. The hyperparameter $\lambda$ controls the strength of emphasis on difficult samples: smaller $\lambda$ assigns higher weight to high-loss (difficult) samples, while larger $\lambda$ approaches uniform weighting (equivalent to standard ERM loss).

---

### Official Review · Reviewer_nhPm · 2025-10-25

**Soundness:** 3
**Presentation:** 3
**Contribution:** 2
**Rating:** 4
**Confidence:** 4

**Summary:**

This paper proposes Uni-VQA, a hybrid VQA framework that uses calibrated confidence scores from a task-specific VQA model to decide when and how to involve an LLM. High-confidence questions are answered locally, low-confidence ones are delegated to the LLM, and intermediate cases use candidate answers for collaboration. The approach improves accuracy while reducing reliance on costly LLM inference.

**Strengths:**

- The related work section is well organized and clearly positions this framework among prior VQA, calibration, and LLM-augmented systems, making it easy to understand its practical motivation.

- The theoretical derivations and more complex details are moved to the appendix, which helps maintain good readability in the main paper.



- The experimental evaluation is comprehensive, including multiple backbones, datasets, and ablation studies.

**Weaknesses:**

- Confidence threshold decisions could use more clarification
  - The routing strategy depends on two confidence thresholds (𝑙,𝑢). It would be helpful to include more explanation on how these values are selected and how sensitive the method is to different threshold settings across datasets or models.

- The effectiveness of TS-VQA candidates may vary depending on the confidence level
  - When the TS-VQA model has low confidence, its proposed candidates may negatively influence the LLM’s reasoning. The current strategy of suppressing candidates only in low-confidence cases is reasonable, but additional analysis on when candidates are beneficial vs. harmful would provide deeper insight into this interaction.

- LLM output reliability is not fully addressed
  - The reliability of LLM outputs is not modeled. Since the LLM handles the most uncertain samples, having some form of uncertainty estimation or error check on the LLM side could further improve the robustness of the overall system.

- Some figures have relatively small, which affects readability. For example, Figure 3 and Figure 14.

- There is no Ethics Statement or Reproducibility Statement in the paper, which are required by the ICLR submission guidelines. Authors may miss this information.

**Questions:**

See weaknesses please

---

> ### Author Response · Authors · 2025-11-24
> **Response to Weaknesses 1 & 2**
>
> ## Response to Weakness 1: Confidence Threshold Decisions
>
> We provide comprehensive answers to both aspects of your question in the revised manuscript:
>
> ### **Part 1. Threshold Selection Process (Appendix F.4):**
>
> Refer to the "Response to Common Questions" on top.
>
> ### **Part 2: Sensitivity Analysis Across Models and Datasets (Appendix G.7):**
>
> The robustness of our threshold selection is extensively analyzed in Appendix Section G.7 through
> cross-model hyperparameter transfer experiments.
>
> **Key Findings are:**
>
> - **Minimal performance degradation:** When applying thresholds tuned for one model to different
> models, the maximum accuracy drop is only 1.24% on COCO-QA, with most deviations below 0.6%
>
> - **Bidirectional robustness:** Transfer performance is symmetric (e.g., ViLBERT→Pythia and
> Pythia→ViLBERT both maintain high accuracy), confirming thresholds are not model-specific
>
> - **Cross-architecture generalization:** Table 9 shows that hyperparameters transfer effectively
> across diverse architectures (e.g., attention-based ViLBERT, transformer-based CLIP-ViL, bottom-up
> Pythia)
>
> **Conclusion:** Our framework does not require careful per-model threshold tuning. The data-driven
> selection process yields robust thresholds that generalize across architectures and maintain near-optimal performance even when transferred, making Uni-VQA practical and scalable for real-world
> deployment.
>
> ---
>
> ## Response to Weakness 2: Effectiveness of TS-VQA candidates
>
> We agree with the reviewer that the usefulness of TS-VQA answer candidates is strongly
> confidence-dependent, and that poorly chosen candidates can mislead the LLM. In fact, this is
> precisely what motivated our dynamic delegation design, and we already perform a detailed analysis
> of when candidates help vs. hurt in Section 3.3 and Appendix G.5 and G.6, summarized in Table 8
> and Figures 4b and 13.
>
> **What we already analyze (and how it answers the comment):**
>
> 1. **LLM accuracy vs. confidence and number of candidates** (Sec. 3.3, Fig. 4b): LLM accuracy within
> TS -VQA confidence bins for different numbers of answer candidates ($K=0,1,2,10$) directly shows:
>    - At high confidences, giving fewer candidates (small $K$) is best; adding many candidates eventually
> hurts.
>    - At moderate confidence, accuracy improves when more candidates are provided.
>    - At the lowest confidence, LLM performance with candidates drops below the LLM-only baseline
> ($K=0$), i.e., candidates are clearly harmful there.
>
> This is exactly why our framework suppresses candidates in the lowest-confidence region and uses a
> confidence-dependent $K(c)$ elsewhere (Fig. 4a).
>
> 2. **Explicit “beneficial vs. harmful” analysis by confidence region** (Appendix G.5, Table 8). To make
> the interaction more concrete, in Appendix G.5 we compare, within each confidence range, three
> systems: TS -VQA, LLM-only, and LLM with TS-VQA candidates (top- 2), for both standard and
> calibrated TS-VQA models. The key findings for calibrated models are:
>    - In Lowest confidence (c ∈ [0, 0.1]) candidates are harmful: the LLM-only consistently achieves the
> highest accuracy, which confirms that *when TS-VQA is very uncertain, its candidates are mostly
> noisy and hurt the LLM’s reasoning*.
>
>    - In moderate confidences candidates generally help: “LLM + candidates” outperforms both TS-VQA
> and LLM-only, showing that when TS-VQA has partial knowledge, its candidate set is informative
> and beneficial for the LLM.
>
>    - In highest confidence In the highest bin, TS-VQA outperforms the LLM (with or without candidates),
> which supports answering high-confidence questions directly with TS-VQA rather than delegating to
> the LLM, saving cost and avoiding degradation.
>
> 3. **Dynamic vs. fixed top-$K$ candidates (Appendix G.6, Fig. 13):**
>
>    - Achieves higher accuracy for any delegation threshold than any fixed $K$.
>    - For a fixed target accuracy, requires lower LLM delegation percentage, meaning fewer LLM calls.
>
> 4. **Qualitative evidence (Appendix H, Fig. 17):**
>    - Illustrates exactly the phenomenon the reviewer mentions: in the lowest-confidence case, TS-VQA’s candidate list misleads the LLM, and only the LLM-only prompt recovers the correct answer; in moderate-confidence cases, the correct answer is within the top-$K$ contextually relevant answers, hence providing more candidates help; in the highest-confidence case, TS-VQA’s answer is already correct and delegating is unnecessary.

---

> ### Author Response · Authors · 2025-11-24
> **Response to Weaknesses 3-5**
>
> ## Response to Weakness 3: LLM output reliability Not Addressed
>
> We appreciate the reviewer’s insightful comment and agree that modeling the reliability of LLM
> outputs is an important direction, especially because the LLM is invoked on the most difficult cases
> in our framework.
>
> **Acknowledgment of Limitation:** We explicitly acknowledge this limitations in the *Limitations
> and Future Works* section (Sec. J): “our approach still lacks the systematic way of providing the well-calibrated uncertainty estimates on the LLM generated answers” and “As uncertainty quantification
> in LLMs remain an ongoing research challenge, we leave the development of more robust LLM
> calibration strategies for future work.”
>
> **Technical Challenges of LLM Uncertainty Quantification** While state-of-the-art approaches
> like *semantic entropy* (Kuhn et al., 2023, Farquhar et al., 2024) have shown promise for LLM
> uncertainty estimation, they introduce significant practical limitations that conflict with our efficiency objectives: They require generating multiple diverse answers per question (typically 10-20
> samples), then clustering semantically equivalent responses. This dramatically increases inference
> cost, which is precisely what our framework aim to reduce. Given that our framework achieves
> 65-79% LLM delegation and reduces latency from 0.53s to 0.18-0.39s (Table 1), incorporating multi-sampling approaches would undermine these core efficiency benefits. Additionally,
> semantic clustering introduces implementation complexity requiring additional NLP models and
> domain-specific hyperparameter tuning.
>
> **Focus and Scope of This Work:**
> The main technical contribution of our paper is not to solve LLM calibration, but rather: (a) to
> calibrate the TS-VQA using a diverse ensemble (Sec. 3.2, App. G.2–G.4), so that confidence reliably
> reflects correctness, and incorrect predictions are shifted into low-confidence regions, and (2) to
> exploit these calibrated confidences for cost-effective delegation and knowledge exchange between
> TS-VQA and the LLM (Sec. 3.3, 4, 5), improving both accuracy and efficiency relative to LLM-only
> and simple cascades.
>
> In other words, we address reliability where we have the most leverage and theoretical control: on
> the TS-VQA model that outputs delegation signal (calibrated confidences) to decide when to call the
> LLM and what information (candidates) to pass.
>
> Systematically developing and benchmarking a full LLM-side uncertainty module, and integrating
> it with delegation logic, would require substantial additional design and experimentation and is
> orthogonal to the core contributions of Uni-VQA. We therefore treat it as important future work
> rather than part of the present scope.
>
> We hope this clarifies that: (1) the lack of explicit LLM-side uncertainty modeling is already acknowledged as a limitation, (2) LLM calibration remains a nontrivial, active research problem, and (3)
> our main focus in this paper is on calibrated TS-VQA confidences and their use for efficient, reliable
> integration with the LLM, while leaving LLM uncertainty modeling as complementary future work
> rather than the central contribution.
>
> ---
> ## Response to Weakness 4: Small Figures
> We have updated the figures with small fonts in the revised version of the paper. (Figures 3 and 17).
>
> ---
> ## Response to Weakness 5: Ethics and Reproducibility Statements
>
> We appreciate the reviewer’s attention to reproducibility. We note that per ICLR guidelines, the Ethics
> and Reproducibility Statements are both encouraged, but optional. Our work does not raise ethical
> concerns that necessitate a dedicated ethics statement (our broader impact discussion in Appendix
> Section I, addresses societal considerations including environmental impact and privacy).
> Nevertheless, to facilitate reproducibility, we have added a dedicated
> Reproducibility Statement to the revised manuscript, including references to our reproducibility
> section in the Appendix, documentation of implementation details, hyperparameters, and code
> availability. We believe this addition strengthens the paper and appreciate the suggestion.

---

> ### Comment · Reviewer_nhPm · 2025-11-25
>
> Thanks for the extensive rebuttal. I have read the authors' rebuttal and the revised manuscript, and I have no further questions.
>
> Therefore, I update my score to 6 and recommend acceptance.

---

> > ### Author Response · Authors · 2025-11-25
> >
> > We thank the reviewer for their time and engagement with our work. We are glad that the revision and rebuttal addressed your concerns, and we appreciate your recommendation for acceptance.

---

### Official Review · Reviewer_H4gC · 2025-10-30

**Soundness:** 3
**Presentation:** 3
**Contribution:** 3
**Rating:** 6
**Confidence:** 3

**Summary:**

The paper proposes Uni-VQA, a method employing an LLM and a task-specific VQA model to efficiently answer questions. The task-specific VQA model is calibrated to provide reliable confidence scores. With the confidence score, the framework whether the answer from the task-specific VQA model should be processed by the LLM. Extensive experiments show the effectiveness of the method.

**Strengths:**

1. The motivation for introducing task-specific VQA model is reasonable and practical.
2. The calibration of task-specific VQA model provide reliable confidence score, enabling the interaction between the two VQA models.
3. Extensive experiments show the effectiveness of the method.

**Weaknesses:**

1. RAG (Retrieval-Augmented Generation) methods are not discussed and compared. The task-specific VQA model serves as a role of providing specific knowledge for LLM, which is similar to external and up-to-date information in RAG methods. The advantages and disadvantages of the proposed method compared to RAG methods should be discussed and compared.
2. The training cost of the task-specific VQA models and the distillation is not reported. It is unclear whether it would be the bottleneck of the framework.

**Questions:**

1. What are the advantages and disadvantages of the proposed method compared to RAG methods?
2. What is the training costs of the task-specific VQA models and the distillation?

---

> ### Author Response · Authors · 2025-11-24
>
> ## Response to Weakness and Question 1
>
> We appreciate the reviewer for pointing out the connection to Retrieval-Augmented Generation
> (RAG). We agree that our framework shares the high-level idea of augmenting LLM with an external
> module. We’ll clarify this connection and their key differences and their complimentary roles.
>
> **Relation to RAG and Key Differences:**
> RAG methods augment an LLM with retrieved textual evidence from a large external corpus (e.g., the
> web, knowledge bases, and documents), and are primarily aimed at mitigating missing or outdated
> LLM knowledge. In contrast, in Uni-VQA, the external component is a trainable TS-VQA model that
> operated directly on the image and question and outputs (1) candidate answers and (2) a calibrated
> confidence score. The LLM is not invoked for every question. It is called only for low- and mid-
> confidence cases, and in the ”consultant” mode it receives a small structured set of TS-VQA answer
> candidates instead of long documents.
> Thus RAG and Uni-VQA share the idea of enriching an LLM with external knowledge, but they target
> different bottlenecks: RAG focuses on external knowledge coverage, whereas Uni-VQA focuses on
> confidence-guided delegation and collaboration between a visual specialist (TS-VQA) and an LLM
> for cost-efficient, reliable VQA.
>
> **Key differences and trade-offs vs RAG:**
> 1. **Primary goal: efficiency-aware collaboration vs. knowledge expansion.** Standard RAG still queries LLM on every question; it changes what context the LLM sees, not
> how often it is invoked. Uni-VQA explicitly optimizes the frequency and mode of LLM usage:
> high-confidence questions are answered by TS-VQA alone, low-confidence ones go to the LLM, and
> mid-confidence ones use TS-VQA candidates. This yields higher accuracy and and better confidence
> reliability than LLM-only while dramatically reducing LLM calls.
> 2. **External module: visual expert vs. knowledge retriever.** In RAG, the external module is a retrieval module, and improvements come from better retrieval/reranking. In Uni-VQA, it is a calibrated TS-VQA, that encodes domain-specific visual
> knowledge.
>
> **Where RAG would help: lowest-confidence / OOD / outside-knowledge regime.**
> We fully agree that RAG is especially usefull when the question requires outside knowledge for the
> TS-VQA. Our calibrated method is designed to push such cases into the lowest confidence bins, so
> they are routed away from TS-VQA and toward the LLM (and, potentially, a RAG-augmented LLM).
> - In Appendix G.11 we evaluate robustness of calibrated confidences under distribution shift using
> the AdVQA OOD dataset. We show that although all models degrade under this OOD setting, our
> calibrated TS-VQA maintains lower ECE than the standard TS-VQA across architectures (Table 15),
> and crucially shifts incorrect predictions toward lowest confidence ranges, rather than remaining
> overconfident in the high confidence ranges (Figs. 15-16).
> - Qualitatively, this means that OOD or “requires world knowledge” questions tend to receive very
> low TS -VQA confidence under our calibration, while in the standard TS-VQA many incorrect OOD
> answers remain overconfident. This behavior is exactly what Uni- VQA needs: such questions are
> automatically delegated to the LLM (teacher mode, with no candidates) and are natural targets for a
> RAG-enhanced LLM.
> To further support this, we evaluated calibrated models trained on VQA-v2 dataset, on the test
> split of OK-VQA dataset (Marino et al., 2019), a knowledge-based VQA benchmark. On the
> OK-VQA test split, in the our calibrated TS-VQA (accuracy: 20%, ECE: 0.11), roughly 56.6%
> of incorrect predictions have confidence below the lower confidence threshold $l$, (as opposed to
> 27% in standard TS-VQA). This indicates that knowledge-heavy questions are much more reliably
> pushed to lowest-confidence region under our diverse ensemble calibration. In Uni-VQA, that region
> is precisely where we do not use TS-VQA candidates and instead let the LLM answer directly.
> Potentially, RAG-augmentation can be employed for lowest confidences (lower than $l$) to improve
> LLM accuracy while avoiding the expensive cost of invoking RAG on every question.
>
> In summary, conceptually, Uni-VQA and RAG are orthogonal while being complementary: one
> controls when and how the LLM is used, the other controls what textual evidence the LLM sees. In
> fact, Uni-VQA could directly wrap a RAG-augmented LLM in its “teacher/consultant” roles without
> changing the TS-VQA calibration or routing logic.
>
> ---
> ## Response to Question 2: Training Costs
>
> Please, see the "Response to Common Questions" on top

---

> > ### Comment · Reviewer_H4gC · 2025-11-26
> >
> > Thanks for the rebuttal. I would encourage the authors to discuss the relation to RAG in the main paper. I keep my original positive rating.

---

> > > ### Author Response · Authors · 2025-11-27
> > >
> > > We thank the reviewer for their time and positive assessment of our work. We have revised the paper to include a discussion of RAG methods. Specifically, we added a discussion to the Related Work section of main paper (Section 2) clarifying the relationship between Uni-VQA and RAG. Additionally, we include a new paragraph at the end of our OOD analysis (Appendix G.11) demonstrating the opportunity for RAG integration in Uni-VQA framework.

---

### Official Review · Reviewer_7cPv · 2025-10-31

**Soundness:** 3
**Presentation:** 3
**Contribution:** 3
**Rating:** 6
**Confidence:** 5

**Summary:**

This paper proposes a novel framework called Uni-VQA for visual question answering, aiming to address challenges in directly applying large language models to VQA, such as suboptimal performance in specialized domains, high computational costs, and lack of uncertainty quantification. The key contributions include: (1) developing a calibration technique based on a diverse ensemble to improve the reliability of confidence estimates for task-specific VQA models; (2) introducing a confidence-guided knowledge exchange mechanism that dynamically decides whether to delegate to an LLM and how to provide candidate answers based on TS-VQA confidence scores, optimizing accuracy and efficiency; and (3) validating the framework through theoretical analysis and extensive experiments on VQA-v2 and COCO-QA datasets, showing superior performance over using LLM or TS-VQA alone while significantly reducing computational overhead. The paper also explores knowledge distillation for faster inference and provides analysis on carbon emissions and latency, highlighting environmental sustainability benefits.

**Strengths:**

The paper novelly integrates confidence-guided mechanisms with LLM-VQA collaboration, differing from traditional model cascades or simple delegation. The dynamic candidate answer selection based on confidence intervals is a creative combination, underexplored in existing work.

The method has theoretical depth and experiments are comprehensive, covering multiple VQA modelsand datasets with consistent results. Ablation studies validate component importance. Writing is concise, and figuresintuitively explain complex concepts, with clear mathematical derivations.

The work directly targets AI scalability and sustainability, reducing LLM carbon footprint, with potential impact on high-stakes applications , aligning with green AI trends.

**Weaknesses:**

While tested on VQA-v2 and COCO-QA, the paper does not include more diverse datasets (e.g., medical VQA or long-tailed distributions), potentially limiting generalizability. Also, LLM usage is limited to Mistral-7B and LLaVA, without extension to larger models , failing to fully assess scale effects.

The calibration technique relies on diverse ensembles, increasing training overhead, and although distillation mitigates this, it may affect deployment ease.

Comparison with recent VQA methods (e.g., Transformer-based variants) is limited; the paper focuses on traditional baselines.

Theorem 4.2 relies on inverse relationship between entropy and confidence, but strict proof in multi-class settings depends on uniform distribution assumptions, which may deviate in practice.

**Questions:**

1.How does Uni-VQA handle modal missingness or distribution shifts? For example, if image quality is poor or questions are out-of-distribution, does confidence calibration remain reliable?

2.The paper mentions that confidence thresholds are determined using a validation set. Could you provide more specifics about the optimization process? What objective function was used to balance accuracy and efficiency during threshold selection?

3.While the paper demonstrates inference efficiency, could you provide more details about the training computational costs of the diverse ensemble approach? How does the training time scale with the number of ensemble members?

4.How does the framework handle concept drift or distribution shifts over time? Have you considered mechanisms for continuous adaptation of the confidence thresholds?

---

> ### Author Response · Authors · 2025-11-24
> **Part 1: Response to (W1) diverse datasets, (W2) deployment concerns, (W3) modern VQA architectures**
>
> ## Response to Weakness 1: Diverse datasets, and larger LLMs
>
> We appreciate reviewer’s comment, and are actively extending our experiments:
>
> **Diverse domains:** A full Medical VQA domain is underway. Our preliminary results on calibration effects of our diverse ensemble on Path-VQA dataset, demonstrates clear improvements on ECE with similar accuracy (reduced from 0.19 to 0.06) compared to the Standard model. We will include the complete Uni-VQA analysis.
>
> **Larger LLMs:** We are extending our LLMs, to demonstrate scale effects. We will include the complete results.
>
> ---
> ## Response to Weakness 2: Deployment Concerns for Diverse Ensemble
>
> We agree that deployment simplicity is important. Crucially, the diverse ensemble is only used offline during training. At deployment, Uni-VQA uses a single distilled TS-VQA model (plus a frozen LLM), making deployment no more complex than using a standard TS-VQA model.
>
> Regarding the training overhead, our ensemble requires roughly 3× the training cost of a single TS-VQA model (15 to 366 GPU hours depending on backbone; see Table 16 in Appendix G.12).
> However, this one-time expense is modest and yields substantial inference savings. To match LLM-
> only accuracy, Uni-VQA delegates only 8-40% of questions to the LLM (Table 3), compared to 100%
> for LLM-only baselines.
>
> Prior work on environmental impact of large AI systems shows that inference often dominates the
> total footprint in deployed systems (Patterson et. al. 2021, Weidinger et. al. 2022; Luccioni et al
> 2024), as discussed in Appendix sec I. Thus, while there is a modest one-time training overhead, the
> deployed system remains simple and delivers significant savings in LLM inference costs and latency.
>
> ---
> ## Response to Weakness 3: Limited Modern VQA Architectures
>
> While our original evaluation included BEiT-3 (Wang et al., 2023), a state-of-the-art transformer-based model, evaluation on a broader range of recent architectures would strengthen our claims and demonstrate generalizability. To this end, we added ViLT (Kim et al., 2021) to demonstrate generalizability across modern architectures.
>
> **Experimental setup:** We trained Calibrated ViLT using the same diverse ensemble methodology, with the same configuration as ViLBERT for both datasets, with ensemble size of 3.
>
> **Results and findings:** Complete evaluation is provided in the Appendix Section G.10 of the
> revised paper. Key results and observations are summarized below:
> 1. Calibration effectiveness transfers to modern architectures: Our diverse ensemble approach reduces
> ECE from 0.17 to 0.02 (COCO-QA) and from 0.21 to 0.07 (VQA-v2), demonstrating substantial
> calibration improvements.
> 2. Uni-VQA maintains superior performance: The integrated framework achieves the highest accuracy
> (76.33% on COCO-QA and 74.31% on VQA-v2), while requiring only 70.47% and 65.31% LLM
> delegation, respectively. Results on ViLT are consistent with patterns observed across other
> architectures.
>
> 3. Method generalizability: The consistent improvements across both traditional and modern transformer-based architectures validate that our calibration and delegation mechanisms are architecture-agnostic and applicable to
> contemporary VQA models.
>
> Complete experimental details and full results are provided in Appendix G.10 (Table 14).
>
> * Kim, Wonjae, Bokyung Son, and Ildoo Kim. ”Vilt: Vision-and-language transformer
> without convolution or region supervision.” International conference on machine learning. PMLR, 2021.

---

> ### Author Response · Authors · 2025-11-24
> **Response to (W4) theorem 4.2**
>
> ## Response to Weakness 4: Theorem 4.2
>
> We appreciate the reviewer’s observation that, in a general multi-class setting, higher entropy does not strictly imply lower maximum probability, (e.g., one can have $H(p) > H(q)$ while also $\max_i p_i >
> \max_i q_i$). Nevertheless, entropy and confidence are tightly linked. In particular, given the number of
> classes $A$, the entropy constrains $c = \max_i p_i$ to lie in an admissible interval that limits the
> region in which the maximum class-probability can fall. More precisely,
> Let $p$ be a distribution over $A ≥ 2$ classes and $c = \max_i p_i$. Define the binary entropy $h(c) = −c \log c−(1−c) \log(1−c)$. Then, $H(p)$ is bounded by:
>
> $$− \log c ≤ H(p) ≤ h(c) + (1 − c) \log(A − 1)$$
>
>
> **proof:**
>
> 1. *Lower bound:* The lower bound is the min-entropy $H_\infty(p)$ (Rényi entropy of order infinity, Rényi 1961), which satisfies $H_\infty(p) = - \log \max_i p_i = - \log c \leq H(p)$. Equivalently:
>
> $$H(p) = - \sum_i p_i \log p_i \geq - \sum_i p_i \log c = - \log c$$
>
> 2. *Upper bound:* The upper bound is achieved when the remaining probability mass $(1 − c)$
> is distributed uniformly. Without loss of generality, suppose the maximum probability is $p_1=c$. Let $r=(r_2,\cdots, r_A)$ be the distribution over the remaining $A-1$ classes. i.e., $p_i = (1-c) r_i$ for $i=2,\cdots, A$ (with $\sum_{i=2}^A r_i = 1$). Then:
>
> $H(p) = −c \log c − \sum_{i=2}^A p_i \log p_i = −c \log c − \sum^A_{i=2} (1 − c)r_i(\log(1 − c) + \log r_i) = −c \log c −
> (1 − c) \log(1 − c) − (1 − c) \sum_{i=2}^A r_i \log r_i = h(c) + (1 − c)H(r)$
>
> Since $r$ is a distribution over $A-1$ classes, we have $H(r) ≤ \log(A − 1)$ (with equality when $r$ is uniform). Thus:
> $$H(p) ≤ h(c) + (1 − c) log(A − 1)$$
>
> Define $F_A(c) = h(c) + (1 − c) \log(A − 1)$. We have:
>
> $F'_A(c) = \log \frac{1−c}{c} − \log(A − 1) = \log \frac{1−c}{c(A−1)}$.
>
> Moreover, by definition we have $c \geq \frac{1}{A}$, and for $c>\frac{1}{A}$:
>
> $$c(A − 1) > 1 − c \Rightarrow \frac{1−c}{c(A−1)} < 1
> \Rightarrow F'_A(c) < 0$$
>
> it follows that $F_A(c)$ is strictly decreasing on $(\frac{1}{A},1]$. Thus $F_A$ is invertible on $[\frac{1}{A},1]$. Let $g_A$ denote its inverse, so that $g_A(F_A(c)) = c$:
>
> $$H(p) ≤ F_A(c) \Rightarrow c ≤ g_A(H(p))$$
>
> The lower
> bound $c ≥ e^{-H(p)}$ can easily be obtained from $- \log c ≤ H(p)$. Hence, we have the following bound
> on the confidence as:
>
> $e^{-H(p)} ≤ c ≤ g_A (H(p))$.
>
> This inequality implies that as entropy $H(p)$ increases, both lower and upper bounds on confidence decrease, so large confidence values become incompatible with high entropy. E.g., for VQA-v2 with $A = 3129$, at $H(p) = 1$, $c ≤ 0.91$, while at $H(p) = 4$, $c ≤ 0.59$. Therefore,
> moderately increased entropies already exclude very high confidences, thereby reducing the over-confidence.
>
> **Empirical relationship between entropy and confidence:**
>
>
> The theoretical bound $e^{−H(p)} ≤ c ≤ g_A (H(p))$ is illustrated in Fig. 9a (Appendix Sec. D), where we plot the empirical
> pairs $(H(p(x)), \max p(x))$ of our model together with the theoretical lower and upper bound curves, demonstrating the decrease in confidence bounds, as entropy increases.
>
> Fig. 9b plots the per-sample changes in entropy and confidences relative to a standard model: A striking $94.1$% of the affected examples lie in the upper left quadrant ($\Delta H>0, \Delta c < 0$), demonstrating that an increase in entropy almost always ($96.72$% of samples with $\Delta H>0$) simultaneously exhibited a decrease in confidence, providing strong empirical support for our theoretical analysis.
>
> Quantitatively, the Pearson correlation between entropy and confidence is $r = −0.9236$, and Spearman rank correlation is $\rho = −0.9604$ over the validation set. Both statistics confirm a pronounced inverse relationship between empirical entropies and confidences.

---

> ### Author Response · Authors · 2025-11-24
>
> ## Response to Question 1: OOD case study
>
> We thank the reviewer for this insightful question. To address robustness under distribution shift, we
> conduct experiments on AdVQA (Sheng et al., 2021), an adversarial out-of-distribution benchmark,
> using models trained only on VQA-v2 (detailed in Appendix G.11).
>
> **Key findings**:
>
> 1. **Calibration advantage persists under distribution shift.** Despite significant accuracy
> drops (e.g., ViLBERT: 66.9%→32.1%), which is expected due to distribution shift, our calibrated
> models maintain substantially better calibration than standard models at comparable accuracy levels
> (e.g., ViLBERT: ECE 0.2 vs 0.37; VisualBERT: ECE 0.14 vs. 0.36):
>
> 2. **Confidence distribution analysis:** By analyzing confidence distributions (Figs. 15-16, Appendix G.11), we observe that even under distribution shift, our calibrated models push
> incorrect predictions toward lower confidence ranges while maintaining higher confidence on correct
> ones. Meanwhile, standard models show severe over-confidence. This mirrors the behavior observed in
> in-distribution scenario (Figs. 11-12).
>    - Improved calibration under distribution shift preserves the effectiveness of confidence-guided
> delegation: more incorrect TS-VQA predictions fall into low-confidence regions and are appropriately
> routed to the LLM (Table 15).
>    - **Increased delegation as distribution-shift detection:** The LLM delegation
> percentages of Uni-VQA increase dramatically under distribution shift (e.g., CLIP-ViL: 65% →
> 99.86%; VisualBERT: 73% to 96.03%; ViLBERT: 67% → 91.76%; see Table 15). This substantial
> increase is precisely an expected and desired behavior, and serves as a natural indicator of distribution
> shift. Rather than producing confidently wrong predictions, our calibrated models appropriately
> express uncertainty and defer to the LLM.
>
> Complete details and results are provided in Appendix G.11 (Table 15, Figs. 15-16).
>
> ---
> ## Responses to Question 2 (Confidence Threshold Determination Strategy), and Question 3 (Diverse Ensemble Training Costs)
> Refer to the ”Response to Common Questions” on top.
>
> ---
> ## Response to Question 4: Adaptation under concept drift
>
> We address this question in three parts: (1) empirical evidence on our framework already exhibits
> desirable behavior under distribution shift, (2) how our threshold-learning procedure naturally
> supports continuous adaptation, and (3) two concrete adaptation scenarios for different drift severities.
>
> **(1) Built-in robustness under distribution shift:** In response to question 2, we evaluated Uni-VQA
> on the AdVQA dataset as an OOD scenario (Appendix G.11), where all models are trained on
> VQA-v2 and tested on adversarial examples. We observed:
>    - *(a) Calibration degrades gracefully:* While all models degrade in accuracy and calibration, our calibrated models maintain substantially lower ECE (better calibration) than standard TS-VQA (Table. 15), shifting more incorrect predictions to
> the lower confidences rather than remaining overconfident (Figs. 15-16).
>    - *(b) Automatic delegation increase:* In response to distribution shift, our framework dramatically increases LLM delegation
> (e.g., CLIP-ViL: from 69% to 99%), because the calibrated TS-VQA produces more low-confidence
> predictions on unfamiliar data patterns, triggering delegation without manual threshold adjustment.
> This demonstrates that Uni-VQA has a **built-in LLM-fallback mechanism:** under
> distribution shift, TS-VQA becomes appropriately uncertain and automatically relies more on the LLM,
> rather than producing confidently wrong answers.
>
> **(2) Threshold learning: Data-driven**: Our confidence thresholds $l$, $u$ and candidate selection
> mechanism $K(c)$ are determined purely from validation data (Appendix F.4). This procedure can
> be periodically re-run on fresh data to adapt thresholds to evolving distribution, without retraining
> the TS-VQA. Our cross-model hyperparameters analysis (Appendix G.7) shows hyperparameters
> are robust, suggesting they don’t need careful and frequent re-tuning unless distribution shift is
> substantial.
>
> **(3) Two adaptation scenarios based on drift severity:** We propose two approaches:
>
>    - *Scenario A: Lightweight threshold adaptation (for gradual drift):* when question styles or visual
> patterns slowly evolve, while TS-VQA remains mostly relevant, threshold learning can be periodically
> re-estimated on a recent labeled validation data, without model retraining.
>    - *Scenario B: TS-VQA Re-training (for substantial drift):* a substantial and sustained increase in
> low-confidence predictions and LLM-delegation (as observed on AdVQA), indicates a dramatic
> distribution shift where the TS-VQA is not relevant anymore. In such case, calibrated TS-VQA can
> be re-trained on new distribution (by collecting labels on high-delegation samples), and re-training
> the calibrated TS-VQA on the augmented datasets, adapting to the new distribution, followed by
> re-estimation of thresholds.

---

> > ### Comment · Reviewer_7cPv · 2025-11-27
> >
> > The author has addressed all my concerns, so I believe the original decision should stand.

---

> > > ### Author Response · Authors · 2025-11-27
> > >
> > > We thank the reviewer for confirming that we have addressed all their concerns, and for their time and valuable insights. The reviewer's detailed feedback has strengthened our revision. We appreciate the positive assessment and support for acceptance.

---

> ### Author Response · Authors · 2025-12-03
> **Update on remaining experiments**
>
> Below we provide the additional experimental evidence on (1) larger LLMs, (2) a medical domain dataset, as promised in our earlier response.
>
> ## Larger LLM Generalization
> To address concerns about generalizability to stronger LLMs, we repeat the same Uni-VQA experiment as in Table 1 of main paper, with **LLaMA-70B (4-bit quantized)** and **Qwen3-14B** LLMs. We use ViLBERT as the TS-VQA, on COCO-QA dataset. We report accuracy as well as LLM-Delegation (\%) (i.e. the fraction of questions delegated to the LLM; lower delegation = cheaper). *Our calibrated ViLBERT TS-VQA achieves accuracy of 70.59 on COCOQA (See Table 1)*
>
> | LLM                   | Variant     | **Accuracy (%)** | **LLM-Delegation (%)** |
> | --------------------- | ----------- | ---------------- | ---------------------- |
> | **LLaMA-70B (4-bit)** | LLM-only    | 71.63            | 100                    |
> |                       | **Uni-VQA** | **76.84**        | **51.32**              |
> | **Qwen-14B**          | LLM-only    | 72.96            | 100                    |
> |                       | **Uni-VQA** | **77.54**        | **67.10**              |
>
> These results mirror the same conclusion as Table 1 (with Mistral-7B as the LLM): Our Uni-VQA continues to improve over **both** TS-VQA-only and LLM-only, while reducing reliance on the LLM, by ~33-49\%.
>
>
> ### **Delegation-Efficiency vs. Naive LLM-VQA (threshold) (i.e. standard TS-VQA+LLM)**
>
> We additionally follow the same analysis done in Table 2, and compare against the naive "Standard+LLM" threshold-delegation baseline (i.e. naive delegation based on *uncalibrated* TS-VQA confidence, without knowledge exchange), with the larger LLMs. In this analysis we ask: *"How much delegation is needed to reach the baseline's best accuracy?"*.
> - Naive LLM-VQA (standard) reaches its maximum accuracy of **73.60** and **74.04** at **29.96\%** LLM-delegation, respectively with LLaMA-70B and Qwen3-14B. Uni-VQA matches/exceeds it, with only **16.4\%** delegation, i.e. the same accuracy, with substantially lower LLM usage.
>
> Consistent with Table 2's takeaway in the main paper, Uni-VQA is not only more accurate, but also more delegation-efficient: at the same target accuracy, it needs meaningfully less LLM invocation.
>
> ---
>
> ## Update on Medical VQA (specialized domain)
>
> To evaluate in a specialized medical domain we used:
> - Dataset: PathVQA
> - VLM: EXGRA-MED: Extended Context Graph Alignment for Medical Vision-Language Models (based on LLaVA-7B) as the large vision-language model.
> - TS-VQA backbone: BAN-based PathVQA baseline models (https://github.com/KaveeshaSilva/PathVQA/tree/main/baselines/method2)
>
> **Calibration performance**: Our calibration reduces calibration error from **0.19** to **0.06** while maintaining similar accuracy to the standard (uncalibrated) model.
>
> **Open-ended accuracy**:
> - VLM-only achieves accuracy of **36.11\%**
> - TS-VQA only (BAN): achieves accuracy of 31.80\%
> - Our Uni-VQA achieves accuracy of **36.64\%** with **44\%** VLM-delegation, surpassing both TS-VQA, and VLM-only.

---

### Author Response · Authors · 2025-11-24
**Response to Common Questions**

We thank all the reviewers for their thoughtful and constructive feedback. Below, we first provide responses to common questions raised by multiple reviewers.

---
# Training Computations Costs (Response to Reviewers 7cPv and H4gC)

**Diverse Ensemble Training and Distillation Costs**:
Detailed training cost analysis is provided in Appendix G.12 (Table 16).

**Key insights** are:
- Our diverse ensemble consists of 3 independently trained TS-VQA models, resulting in 3× the
training cost of a single standard model.
- Across backbones, ensemble training requires 15-366 GPU hours on A100 GPUs (e.g., Pythia: 15
hrs; CLIP-ViL: 366 hrs; BEiT-3: 216 hrs).
- **Scalability:** Training time scales linearly with ensemble size E, as members are trained
independently. Total training time = E× (single model training time).
- **Distillation: (Appendix G.9, Tables 11-12)** optionally adds ∼ 1/3 of ensemble training time but
reduces TS-VQA inference latency by up to 60% while preserving both accuracy and calibration.

While ensemble training incurs a 3× overhead, this is a one-time expense that is modest compared to
LLM training/finetuning and inference costs and is quickly amortized by recurring inference savings.
As detailed in Appendix G.12 (Table 17), Uni-VQA achieves 1.34-6.64× inference speedup over
LLM-only baselines depending on the backbone, making the upfront training investment worthwhile
for large-scale deployments.

---
# Confidence Threshold Determination (Response to Reviewers 7cPv and nhPm)

The thresholds $l$, $u$, and the confidence-dependent candidate function $K(c)$ are chosen in a fully
data-driven way on a held-out validation set. The process has two steps:

1. **Per-confidence-bin policy selection (optimize accuracy)**. We first partition the calibrated
TS-VQA confidence range into bins (e.g. 0-0.1, 0.1-0.2, ...). For each bin we evaluate several
interaction modes on the validation set:
   - TS-VQA only (no LLM),
   - LLM only (no candidates),
   - LLM with top-K candidates from TS-VQA, for different K (we limit $K$ to {1,2,5,7,10}).

In each confidence bin we simply pick the model that achieves the highest VQA accuracy on
validation. This gives a discrete mapping “confidence range → best model.”

2. **Deriving thresholds and a smooth $K(c)$.** From this mapping we then:
   - Define the low-confidence threshold $l$ as the highest confidence range where LLM-only is best.
   - Define the high-confidence threshold $u$ as the lowest confidence range where TS-VQA-only is best.
   - In the intermediate region $[l, u]$, for each bin, we find the $K$ performing best, and fit a simple
monotone function $K(c)$ so that lower confidence gets more candidates and higher confidence gets
fewer.

Thus, the explicit objective on the validation set is *accuracy* in each confidence region; efficiency
is enforced structurally by the upper confidence threshold $u$:
- We do not invoke the LLM where TS-VQA alone is already most accurate (high confidence).

As shown in Figs 6-7 and Appendix G.6-G.7, this data-driven policy yields strictly better accuracy
delegation trade-offs than threshold baselines at all delegation levels.

We now include the complete confidence-threshold determination process in Appendix F.4 of the
revised paper.

---

### Author Response · Authors · 2025-12-03
**Rebuttal Summary**

Dear AC,

We thank you for overseeing the review process and the reviewers for their valuable feedback. Three reviewers (7cPv, H4gC and nhPm) explicitly confirmed in discussion that our rebuttal and revision fully addressed their concerns, and their updated evaluations reflect that the paper is now both clearer and stronger. The remaining negative review (FgQE) appears to stem from a *fundamental misunderstanding of our method’s core contribution*, while overlooking the calibration mechanism that is central to our approach and explicitly recognized as a key strength by the other reviewers. Additionally reviewer FgQE's review contains factual inaccuracies (e.g., characterizing our data-driven thresholds as "hard-coded", and requesting comparisons already present in the paper).

We believe our revision, additional experiments and clarifications during rebuttal period have significantly strengthened the paper. Below, we summarize how the rebuttal and revised paper addresses the reviewers' concerns, and summarize what was already in the paper but overlooked.

---

## Reviewer concerns already satisfied in the original paper:
Several reviewers raised concerns about missing content that was already present in our original submission:

**1. Training Computation Costs (7cPv & H4gC)**: Appendix G.12, Table 16 provided complete training costs (15-366 GPU hours) and inference latency breakdowns. We summarized this information in our rebuttal for clarity.

**2. Sensitivity Analysis of Hyperparameters (nhPm, W1):** Appendix G.7 extensively analyzed threshold robustness through cross-model hyperparameter transfer, showing maximum accuracy drop of only 1.24% when transferring parameters across architectures.This analysis demonstrates that our framework is not sensitive to careful tuning of thresholds.

**3. Effectiveness of TS-VQA Candidates (nhPm, W2)**:  Appendices G.5-G.6 comprehensively analyzed when candidates help vs. harm: (a) Table 8 showed candidates are harmful at lowest confidence (LLM-only achieves best accuracy), beneficial at moderate confidence (LLM+candidates outperforms both), and unnecessary at highest confidence (TS-VQA alone is best); (b) Figure 13 demonstrated our dynamic approach strictly dominates all fixed-K variants; (c) Figure 17 provided qualitative examples illustrating this pattern.

**4. Ensemble vs. Distilled Model Comparison (FgQE, W2) :** Appendix G.9, Tables 11-13 explicitly compared ensemble and distilled models within the Uni-VQA framework, showing distilled models maintain ensemble accuracy/calibration while reducing inference latency by 60% and achieving 5-6% better delegation efficiency, directly addressing the reliability-latency tradeoff.

---
## New Experiments and Revisions Made During Rebuttal:
**1. Clarification of Threshold Determination (7cPv, nhPm)**: We added complete data-driven threshold selection methodology in Appendix F.4, detailing our process.

**2. Diverse Datasets and Larger LLMs (7cPv, W1)**:  We added:
   -  *Medical domain:* PathVQA (a medical domain VQA dataset on pathology images) experiments with LLaVA-7B-based medical VLM showing consistent calibration improvements (ECE: 0.19→0.06) and accuracy gains (36.64% vs. 36.11% VLM-only, 31.80% TS-VQA-only) with 44% delegation.
   - *Larger LLMs:* Experiments with LLaMA-70B (4-bit quantized) and Qwen3-14B on COCOQA with ViLBERT, showing same patterns as Tables 1-2: Uni-VQA achieves 76.84-77.54% accuracy (vs. 71.63-72.96% LLM-only) while reducing delegation by 49\% and 33\% respectively with LLaMA-70B and Qwen3-14B.

**3. Modern VQA Architectures (7cPv, W3)**: We added experiments with ViLT (a modern transformer-based vision-language architecture for VQA) to Appendix G.10 (Table 14). The results mirror core findings on other VQA models, showing generalizability across diverse traditional and modern VQA architectures.

**4. Deployment Concerns Clarification (7cPv, W2)**: We clarified that diverse ensemble is only used offline during training. At deployment, Uni-VQA uses a single distilled TS-VQA model and a frozen LLM. While ensemble training requires 3× cost of single TS-VQA (15-366 GPU hours), this one-time expense is modest compared to ongoing inference savings. At 10M queries/day, Uni-VQA saves 11,000-35,000 GPU-hours monthly compared to. LLM-only. Prior work (Patterson et al. 2021; Luccioni et al. 2024) shows inference dominates deployed system footprint over training costs, making our approach substantially more sustainable.

**5. Theorem 4.2 Uniform Distribution Assumption (7cPv, W4):** We added rigorous proof showing entropy mathematically constrains confidence to an admissible interval. Thus, as entropy increases, both bounds decrease, excluding high confidences. Our, empirical validation shows 94.1% of samples show inverse entropy-confidence relationship (Figure 9); Pearson correlation: -0.82, and Spearman correlation: -0.88 show strong negative correlation between confidence and entropy.

---

### Author Response · Authors · 2025-12-03
**Rebuttal Summary (continue: part 2)**

**6. Distribution Shift Robustness (7cPv, Q1):** We added comprehensive OOD evaluation, on AdVQA dataset (adversarial OOD) with models trained only on VQA-v2 (Appendix G.11: Table 15, Figs 15-16). Key findings: (1) *Calibration effect of our diverse ensemble persists under OOD:* Our calibrated models maintain substantially better calibration than standard models (e.g., ViLBERT ECE: 0.20 vs. 0.37); (2) *Automatic delegation increase:* LLM delegation dramatically increases under OOD (e.g., CLIP-ViL: 69%→99.86%), serving as natural distribution shift indicator; (3) *Confidence distributions:* Our models appropriately assign lower confidences to OOD samples rather than producing confidently wrong predictions (Figs 15-16). Additional OK-VQA analysis shows 56.6% of incorrect predictions fall below lower threshold (vs. 27% for standard), confirming knowledge-intensive questions naturally route to low-confidence region.

**7. Adaptation Under Concept Drift (7cPv, Q4)**: We added detailed discussion on adaptation strategy building on the above OOD findings: (1) *Built-in robustness:* Uni-VQA automatically increases delegation under drift without manual intervention, enabled by our calibrated TS-VQA; (2) *Lightweight threshold adaptation strategy:* For gradual drift, periodically re-estimate thresholds on recent validation data without model retraining; (3) *TS-VQA retraining:* For substantial drift (indicated by sustained high delegation), retrain calibrated TS-VQA on augmented dataset and re-estimate thresholds.

**8. RAG Integration and Positioning (H4gC, W1, Q1):** We added an extended discussion in Related Works (Section 2), clarifying: differences between RAG and Uni-VQA and how they are orthogonal and complementary: RAG controls what textual evidence LLM sees; Uni-VQA controls when/how LLM is used. RAG generally invokes LLM on every query; Uni-VQA optimizes LLM invocation frequency (high-confidence → TS-VQA only; low-confidence → LLM; moderate → LLM+candidates). Evidenced by OOD analysis, our diverse ensemble naturally pushes OOD/knowledge-intensive questions to lowest-confidence region (56.6% on OK-VQA), precisely where RAG augmentation would be most beneficial. Uni-VQA could directly wrap a RAG-enhanced LLM for such low-confidence queries without modifying calibration/routing logic.

**9. LLM Output Reliability (nhPm, W3):** This is explicitly acknowledged in our Section J. In response, we additionally clarified why this is beyond our scope: (1) Technical challenge: State-of-the-art approaches (e.g. semantic entropy) require 10-20 LLM responses per question to estimate LLM uncertainty, dramatically increasing inference cost, which directly contradicts our efficiency objectives; (2) Focus of this work: Our contribution is calibrating TS-VQA confidence for reliable delegation decisions, not solving the orthogonal problem of LLM uncertainty quantification; (3) Active research area: LLM calibration remains an ongoing challenge in the literature. We treat this as important future work complementary to our core contributions.

**10. Sustainability Accounting (FgQE, W4):** Figure 1 reported per-query latency and carbon emissions; Table 1 \& 17 showed latency comparisons; Appendix G.12 (Table 16) detailed training costs (15-366 GPU hours).

*Clarified in response:* We explained our sustainability claims focus on reducing inference-time LLM reliance, which dominates deployed system energy use. We added explicit accounting: (a) Table 17 computing effective end-to-end inference latency (115-397ms vs. 534ms LLM-only, representing 1.34-6.64× speedup); (b) Deployment-scale calculation showing at 10M queries/day, Uni-VQA saves 11,000-35,000 GPU-hours monthly vs. LLM-only; (c) Scope clarification referencing Green AI work (Schwartz et al. 2020; Patterson et al. 2021; Luccioni et al. 2024) showing inference dominates deployed footprint. Our results consistently show Uni-VQA requires substantially fewer LLM calls at matched accuracy: the key dimension for sustainability in production VQA systems.

**11. Diverse Ensemble Clarity (FgQE, W5, Q1):** We used the extra allowed page to expand Section 3.2 with details previously provided only in appendix, including explicit DRO weight formulation and explanation of how $\lambda$ controls sample weighting during training, improving main paper clarity.

---
## Minor Concerns

**1. Small figures (nhPm, W4):** We updated Figures 3 and 17 with larger fonts in revised paper.

**2. Reproducibility statement (nhPm, W5):** We added dedicated Reproducibility Statement referencing appendix sections on reproducibility related details: implementation details, hyperparameters, and code availability.

---

### Author Response · Authors · 2025-12-03
**Rebuttal Summary (continue: part 3):  Concerns About Reviewer FgQE's Assessment**

## Concerns About Reviewer FgQE's Assessment (Reject):

We respectfully note that Reviewer FgQE did not engage with our rebuttal despite our comprehensive responses. We wish to bring to your attention several significant issues with this review:

## **1. Fundamental Mischaracterization (W1: "Modest Novelty"):**
Reviewer describes our work as *"essentially confidence-controlled routing,"* which completely misses our core technical contribution: i.e. *the calibration mechanism and its central role in Uni-VQA's effectiveness*. This characterization is contradicted by the other reviewers:

   - 7cPv (Strength): "paper novelly integrates confidence-guided mechanisms with LLM-VQA collaboration, differing from traditional model cascades or simple delegation. The dynamic candidate answer selection based on confidence intervals is a creative combination, underexplored in existing work"

   - H4gC (Strength): "The calibration of task-specific VQA model provide reliable confidence score, enabling the interaction between the two VQA models"

reviewer describes our method as "simple confidence-controlled routing", completely missing our core contribution: the calibration mechanism via diverse ensemble (Section 3.2), and the role of knowledge exchange (Section 3.3).

**Our contributions are:**

- Diverse ensemble calibration via DRO with theoretical guarantees (Lemma 4.1: DE loss upper-bounds regularized cross-entropy; Theorem 4.2: shifts incorrect predictions to low-confidence)
- Dynamic knowledge-exchange through confidence-dependent candidate selection K(c) (Equation 2)
- Three-role integration (not binary routing): TS-VQA only / LLM-only / LLM+candidates

We explicitly compare against simple confidence-threshold routing (Table 2 & 4, Figures 6-7), demonstrating 1-20% delegation reduction at equal accuracy. For example, to achieve 70.07% accuracy with Pythia, threshold-based delegation requires 64.38% LLM usage vs. our 50.06% (14.32% reduction).

## **2. "Hard-Coded Routing" Claim (W3):**
Reviewer FgQE characterizes our delegation as "hard-coded", which is factually incorrect. Our thresholds $l$, $u$ and dynamic candidate selection function $K(c)$ are learned from validation data to maximize accuracy (detailed methodology in Appendix F.4). Our cross-model transfer analysis (Table 9) shows these learned parameters generalize robustly (maximum 1.24% accuracy drop), demonstrating they capture underlying data properties rather than being arbitrary manual choices. This mischaracterization suggests misunderstanding of our data-driven approach.

## **3. Overlooked Comparisons Already in Paper:**
As noted above, Reviewer FgQE requested "ensemble vs. distilled model comparison" (W2), but Tables 11-13 in Appendix G.9 already provided exactly this analysis, showing distilled models maintain performance while improving efficiency. This suggests insufficient engagement with the appendix.

---

### Meta-Review · Area_Chair_VC9o · 2026-01-04

**Summary:**

This paper proposes Uni-VQA, a confidence-aware framework for integrating task-specific VQA models with large language / vision-language models. The central idea is to calibrate the task-specific model’s confidence using a diverse ensemble, and then use these calibrated confidences to guide when and how to involve the LLM, including a non-trivial knowledge-exchange mode where candidate answers are selectively provided.

Reviewers generally agreed that the paper is well-motivated and technically sound, with a strong empirical evaluation. The main concerns raised during review related to (i) the perceived novelty relative to confidence-based routing, (ii) clarity and justification of the confidence thresholds and routing strategy, (iii) training cost and sustainability accounting, (iv) comparison to related paradigms such as RAG, and (v) robustness under distribution shift.

**Reviewer Concerns:**

**Concerns addressed by the rebuttal and revision:**

* **Confidence thresholding and routing strategy:**
  Multiple reviewers asked for clarification on how thresholds and candidate selection are determined and whether they are robust. The authors provided a clear, data-driven validation-based procedure, extensive sensitivity analysis, and cross-model transfer experiments, demonstrating robustness and removing ambiguity.
* **Effectiveness of candidate answers and knowledge exchange:**
  Concerns about when TS-VQA candidates help or hurt the LLM were addressed with detailed per-confidence analyses, ablations, and qualitative examples. The revised paper clearly shows that candidates are beneficial only in intermediate-confidence regimes, motivating the dynamic design.
* **Training cost, efficiency, and sustainability claims:**
  The authors added explicit accounting of training and inference GPU hours, delegation rates, and effective end-to-end latency, clarifying that the ensemble is used only offline and that deployment relies on a distilled model. This adequately supports the efficiency and sustainability claims.
* **Relation to RAG and alternative paradigms:**
  The revised manuscript explicitly discusses the relationship to RAG, clarifying conceptual differences and complementarity, and positioning Uni-VQA as orthogonal rather than redundant.
* **Robustness under distribution shift:**
  Additional OOD experiments and analysis demonstrate that the calibration mechanism remains effective and that delegation to the LLM increases appropriately under shift.


**Outstanding concerns:**

* One reviewer remained unconvinced about the level of novelty, characterising the approach as confidence-controlled routing. This concern appears to stem from a high-level interpretation of the method and was not updated after the rebuttal, despite the added analyses and clarifications.

**Reviewer Scores:**

Reviewer 7cPv: Maintained a positive score throughout and explicitly confirmed that all concerns were addressed after the rebuttal.
Reviewer H4gC: Maintained a positive score; minor concerns were resolved in the revision.
Reviewer nhPm: Updated their score from 4 to 6 after reading the rebuttal and revised manuscript, recommending acceptance.
Reviewer FgQE: Maintained a score of 2 and did not revise their assessment or engage substantively with the rebuttal.

---

### Decision · Program_Chairs · 2026-01-26

Accept (Poster)